# Tubular insulin-induced gene 1 deficiency promotes NAD$^+$ consumption and exacerbates kidney fibrosis

Shumin Li [1,2,3], Jun Qin [1,2,3,4], Yingying Zhao [1,2,3], Jiali Wang [5], Songming Huang [1,2,3]✉ & Xiaowen Yu [1,2,3]✉

## Abstract

**Profibrotic proximal tubules (PT) were identified as a unique phenotype of proximal tubule cells (PTCs) in renal fibrosis by single-cell RNA sequencing (scRNA-seq). Controlling the process of renal fibrosis requires understanding how to manage the S1 subset's branch to the S3 subset rather than to the profibrotic PT subset. Insulin-induced gene 1 (Insig1) is one of the branch-dependent genes involved in controlling this process, although its role in renal fibrosis is unknown. Here, we discovered that tubular Insig1 deficiency, rather than fibroblast Insig1 deficiency, plays a detrimental role in the pathogenesis of renal fibrosis in vivo and in vitro. Overexpression of Insig1 profoundly inhibited renal fibrosis. Mechanistically, Insig1 deletion in PTCs boosted SREBP1 nuclear localization, increasing Aldh1a1 transcriptional activity, causing excessive NAD$^+$ consumption and ER enlargement, as well as accelerating renal fibrosis. We also identified nicardipine as a selective inhibitor of Aldh1a1, which could restore NAD$^+$ and maintain ER homeostasis, as well as improve renal fibrosis. Together, our findings support tubular Insig1 as a new therapeutic target for chronic kidney disease (CKD).**

**Keywords** Insig1; Aldh1a1; NAD$^+$ Metaboloism; Endoplasmic Reticulum Expansion; Chronic Kidney Disease
**Subject Category** Urogenital System

## Introduction

Renal fibrosis is a common outcome of CKD and leads to the final and irreversible loss of renal function (Zeisberg and Neilson, 2010; Zeisberg et al, 2001). CKD is a leading cause of death, however few effective treatments are available (Fearn and Sheerin, 2015). For the identification of new therapeutic strategies and improved management of this chronic condition, a better understanding of the mechanism that mediates renal fibrosis is critical.

Differentially expressed genes (DEGs) have been found to be enriched in metabolic pathways and to be altered in PTCs, as indicated by comprehensive genome-wide human kidney tissue transcriptomics, revealing PTC dysfunction and loss in renal fibrosis (Dhillon et al, 2021). PTCs constitute the most common type of cell in the kidney and are essential for maintaining healthy renal function and preventing renal fibrosis following injury (Chevalier, 2016). The clustering of PTCs involved in unilateral ureteral obstruction (UUO) has led to the identification of six PT cell types, namely, precursor PT, S1, S2, and S3 subsets; proliferating PT subset, and profibrotic PT subset, as determined by the GSE182256 scRNA-seq dataset (Data ref: Doke et al, 2022). The S1 subset can branch out into the S3 and profibrotic PT subsets. During this crucial phase, branch-dependent genes are crucial and notably. Insig1 was included among the branch-specific genes identified.

Insig1, an intrinsic protein in the endoplasmic reticulum (ER), is crucial for lipid and cholesterol synthesis (Azzu et al, 2021; Gong et al, 2006). By interacting with the sterol regulatory element-binding protein (SREBP) cleavage-activating protein (SCAP) and ER enzyme 3-hydroxy-3-methylglutaryl coenzyme A (HMG CoA) reductase, Insig1 controls the proteolytic processing and transcriptional activation of SREBP1 (Feramisco et al, 2004; Yang et al, 2002). SREBP1 transcript levels affect a variety of biological processes, such as tumor development (Xu et al, 2020; Zhang et al, 2022), immunity (Assmann et al, 2017; Luo et al, 2023), and lipid metabolism (Shimano and Sato, 2017; Sundqvist et al, 2005). Recent evidence indicates that Insig1 expression is downregulated in kidney diseases such diabetic kidney disease (DKD) and renal mass loss (Kim et al, 2009; Liu et al, 2022), but no study has been done to determine the effect of Insig1 deficiency in the pathophysiology of CKD or PTCs fibrosis.

Before identifying the role of tubular Insig1 in CKD, we performed a bulk RNA-seq analysis to assess the DEGs in PTCs between Insig1$^{\Delta Kap}$ and Insig1$^{flox/flox}$ mice. Among that in the discovered DEGs, aldehyde dehydrogenases 1 family member A1 (Aldh1a1) was chosen as the downstream target of Insig1 in PTCs. Aldh1a1 is a cytosolic enzyme that catalyses mainly the oxidation of acetaldehyde into acetic acid. It is also involved in retinol metabolism, converting retinol to retinoic acid (Bowles et al,

[1]Department of Nephrology, State Key Laboratory of Reproductive Medicine, Children's Hospital of Nanjing Medical University, 210008 Nanjing, China. [2]Nanjing Key Laboratory of Pediatrics, Children's Hospital of Nanjing Medical University, 210008 Nanjing, China. [3]Jiangsu Key Laboratory of Pediatrics, Nanjing Medical University, 210029 Nanjing, China. [4]Department of Pediatrics, Yancheng City No.1 People's Hospital, 224005 Yancheng, China. [5]Department of Hematology and Oncology, Children's Hospital of Nanjing Medical University, 210008 Nanjing, China. ✉E-mail: smhuang@njmu.edu.cn; yuxiaowen@njmu.edu.cn

© The Author(s)                    EMBO Molecular Medicine   Volume 16 | July 2024 | 1675–1703   **1675**

2016). Recent studies have demonstrated that Aldh1a1 is involved in diseases related to inflammation and metabolism. Overexpression of Aldh1a1 has been linked to obesity, diabetes, cancer, and other illnesses (Haenisch et al, 2021; Lerner et al, 2021; Yue et al, 2022). Furthermore, Aldh1a1 leverages nicotinamide adenine dinucleotide ($NAD^+$) as a coenzyme to oxidize aldehydes into carboxylic acids through reactions that are coupled to the reduction of $NAD^+$ to NADH (Wang et al, 2017).

$NAD^+$ is a substrate of key metabolic enzymes involved in energy metabolism, which includes key cellular processes such as glycolysis, the Krebs cycle, and other metabolic reactions in cells (Canto et al, 2015; McReynolds et al, 2020). Many of $NAD^+$ cellular functions are realized through its role as a cosubstrate of sirtuins (SIRTs), poly (ADP-ribose) polymerases (PARPs), CD38 (Morevati et al, 2022), and Aldh1a1. Depletion of $NAD^+$ leads to numerous inherited and acquired diseases in humans, especially age-related diseases, including neurodegeneration, diabetes, and cancer (Katsyuba et al, 2020; Ralto et al, 2020; Zapata-Perez et al, 2021). In addition, $NAD^+$ depletion can lead to ER expansion mediated by the deribosylation of the ER stress sensor GRP78/BiP, causing it to be cleaved off from kinase/endoribonuclease inositol-requiring enzyme 1α (IRE1α) (Wu et al, 2021). In acute kidney injury (AKI) and CKD, substantial decreases in the $NAD^+$ level disrupt energy generation. Increasing evidence indicates that increases in the $NAD^+$ level may protect PTCs from a variety of acute stressors, such as ischemia–reperfusion, toxicity and systemic inflammation (Clark et al, 2022; Ralto et al, 2020). $NAD^+$-dependent maintenance of PTC metabolic health may also attenuate long-term profibrotic responses that can lead to CKD (Takahashi et al, 2022).

Due to the important roles of Aldh1a1 in maintaining $NAD^+$ homeostasis in PTCs, compounds that modulate Aldh1a1 activity must be identified. In the present study, nicardipine was identified as a selective Aldh1a1 antagonist. Nicardipine is a dihydropyridine-type $Ca^{2+}$ channel blocker (CCB) that exhibits unusual cerebrovascular characteristics and high antihypertensive action (Amenta et al, 2009). However, whether nicardipine affects renal fibrosis or Aldh1a1 deactivation and the related mechanisms are unknown.

In this study, we investigated tubular Insig1 influences on CKD progression and sought to determine whether Aldh1a1 plays a role in mediating the effects of Insig1. We used PTCs-specific and fibroblast-specific Insig1-knockout mice and whole-body Aldh1a1-knockout mice subjected to UUO or 5/6 nephrectomy (5/6 Nx) to evaluate the importance of PTC-specific Insig1 in conferred protection against renal fibrosis and the maintenance of $NAD^+$ homeostasis. The findings suggest potential candidate targets and drugs for treating CKD.

## Results

### The downregulation of Insig1 was associated with CKD

To identify the co-occurrences with the fibrosis of PTCs, we analyzed the scRNA-seq dataset GSE182256 which was obtained from the GEO database. Following a quality control process, the main kidney cell types were clustered (Fig. EV1A,B), and the PTCs were reclustered to identify PT subsets based on canonical markers from the original publication. Uniform manifold approximation and projection (UMAP) allowed two-dimensional visualization of the main cell types

and the PT subset (Figs. 1A and EV1C). The pseudo-time trajectory intriguingly revealed that S1 subset were transformed into S3 and profibrotic PT subgroups (Fig. 1B). A heatmap showed that after UUO, the profibrotic PT subgroup exhibited the highest level of ER stress of all the PT subsets (Fig. 1C). Then, in kidney biopsy samples from patients with CKD, we observed an apparent ER expansion and vesiculation in PTCs using an electron microscope and immunofluorescence (IF) staining for the ER marker calreticulin (Fig. 1D). In addition, PTCs from mice with UUO and 5/6 Nx-induced CKD displayed comparable ER alterations and increased ER stress (Figs. 1E and EV1D,E). According to these results, the pathophysiological pathways underlying PT fibrosis involved ER stress.

To identify the genes that may prevent the transition of the S1 subset into the profibrotic PT subset, we applied the BEAM technique to branch point 2. The expression of these genes along with the pseudo-time were shown in the heatmap (Fig. 1F). Then, we employed hdWGCNA to cluster branch-dependent genes. Branch point 2 involved three coexpressed gene modules (Fig. 1G). Three gene modules' eigengene levels from four PT subgroups were shown in the boxplot. We chose the blue module after comparing the gene expression variations between the S1 subset transiting into the S3 subset and the S1 subset transiting into profibrotic PT subset. Compared to those in the S2 subset, the eigengene levels gradually increased in the S1 and S3 subsets but decreased in the profibrotic PT subset (Fig. 1H). There were 120 genes in the blue module, and to identify target genes, we compared these genes with genes that were downregulated in both the GSE118341 dataset and our bulk RNA-seq dataset (Fig. 1I). Next, we combined GO analysis of these 17 similar genes with ER stress-related genes. Insig1 was the only gene identified in the ER unfolded protein response (Fig. 1J).

Insig1 was widely expressed in PTCs (Figs. 1K and EV1F) and with its expression was reduced in the UUO group (Figs. 1L and EV1G–K). In addition, we used an online database (GSE145053) to confirm the decrease in Insig1 mRNA levels in UUO models (Data ref: Conway et al, 2020) (Fig. 1M). Consistent with its alteration in the UUO mouse model, Insig1 was downregulated in kidney biopsies of children with kidney disease compared to controls (Fig. 1N,O). Table EV1 presents a list of the clinical details of children with kidney disease. In addition, the expression of Insig1 in "Ju CKD TubInt" dataset (Data ref: Berthier et al, 2012; Ju et al, 2013; Ju et al, 2015; Reich et al, 2010) showed a strong positive correlation with the estimated glomerular filtration rate (eGFR), but showed a negative correlation with the level of serum creatinine (SCr) (Fig. 1P). These results demonstrated that the downregulation of Insig1 in CKD may contribute to the progression of renal fibrosis.

### Insig1 deficiency in PTCs aggravated UUO-induced CKD

Next, we produced a PTCs-specific Insig1-knockout (Insig1$^{ΔKap}$) mice (Fig. 2A). The protein expression in isolated renal tubule tissue confirmed the reduction in Insig1 expression in Insig1$^{ΔKap}$ mice compared to Insig1$^{flox/flox}$ littermates without Cre recombinase activity (Fig. 2B,C). To elucidate the role of Insig1 in CKD renal fibrosis, we subjected Insig1$^{ΔKap}$ and Insig1$^{flox/flox}$ mice to UUO for 14 days to generate classical renal fibrosis model mice. The Insig1$^{ΔKap}$ mice showed worsened collagen and Fibronectin (FN) deposition in the kidney compared to that in the Insig1$^{flox/flox}$ mice, as determined by Sirius Red staining and Immunohistochemical (IHC) staining

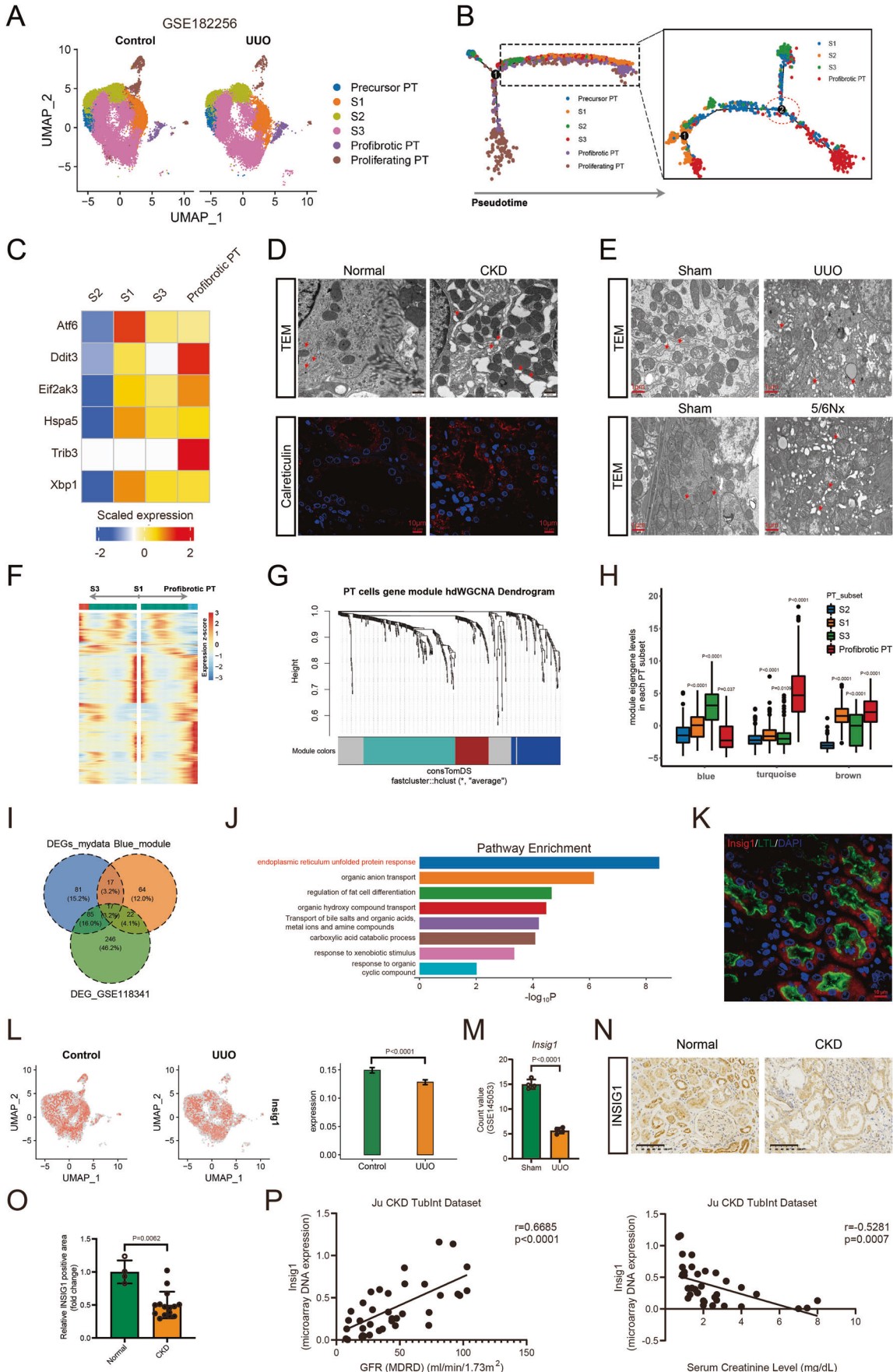

◄ **Figure 1. The downregulation of Insig1 was associated with CKD.**

(A) Subclustering analysis of PTCs in Control and UUO kidneys (*n* = 6 in Control group; *n* = 2 in UUO group). (B) Cell trajectory analysis via Monocle with a heatmap showing changes in gene expression along the trajectory. (C) Heatmap analysis of ER stress in each PT subset. (D) Representative electron micrographs of the ER in PTCs (children with kidney disease, scale bars, 500 nm; red arrows denote expanded ER) and calreticulin in the kidneys of children with kidney disease (scale bars, 10 μm; *n* = 4 in Normal group; *n* = 9 in CKD group, biological replicates). (E) Representative electron micrographs of the ER in PTCs (scale bars, 1 μm; red arrows denote expanded ER; *n* = 3 in each group, biological replicates). (F) The expression trends of branch-dependent genes along with the evolutionary trajectory are shown in a heatmap. (G) Gene modules of branch-dependent genes were clustered by hdWGCNA. (H) Comparison of module eigengene levels in each PT subset (the upper and lower edges of the box are the upper and lower quartiles, the center line of the box is the median, the box whisker boundary is the maximum and minimum, and the scatter point is the outlier). (I) Genes common to the blue gene module, the DEG set in the GSE118341 dataset and the DEG set of our UUO experimental dataset. (J) The enriched biological functions of 17 common genes and 6 typical ER-related genes. (K) Representative images of IF (Insig1) in the normal kidneys (scale bars, 10 μm; red: Insig1; green: LTL; blue: DAPI; *n* = 6 in each group, biological replicates). (L) scRNA-seq analysis of Insig1 in Control and UUO groups (*n* = 6 in Control group; *n* = 2 in UUO group). (M) RNA-seq analysis of Insig1 in the Sham and UUO groups (*n* = 4 in each group, biological replicates). (N, O) Representative images and quantification of IHC (INSIG1) staining in the kidneys of children with kidney disease (scale bars, 100 μm; *n* = 4 in Normal group; *n* = 15 in CKD group, biological replicates). (P) Pearson's correlation of the Insig1 level with the GFR and SCr level (GFR, *n* = 38; SCr, *n* = 38). Data information: In (H, L, M), Data are represented as mean ± SD. Student's *t* test. (O) Data are represented as mean ± SD. Mann–Whitney test. (J) Well-adopted hypergeometric test and Benjamini–Hochberg *P* value correction algorithm. Source data are available online for this figure.

(Fig. 2D). Following UUO, the kidneys of Insig1^ΔKap animals displayed considerably lower expression of E-CADHERIN and significantly higher expressions of the fibrotic markers (*Fn*, α-SMA (*Acta2*), COL-I, *Postn* and VIMENTIN (*Vim*)) (Figs. 2E,F and EV2A), as well as higher expression of genes (*Pdgfb*, *Cd74*) representative of the profibrotic PT subset and lower expression of genes (*Slc6a13*, *Slc7a13*, *Cyp4b1*) representative of the S3 subset (Figs. 2G and EV2B). In addition, Insig1 deletion in PTCs increased expressions of ER stress markers (Calreticulin and *Atf6*) and exacerbated UUO-induced ER expansion and vesiculation (Fig. 2H–J), while exerting no effect on the inflammatory response (Fig. EV2C). These results revealed that Insig1 deficiency in PTCs aggravated UUO-induced renal fibrosis and ER stress.

## Insig1 deficiency in PTCs aggravated 5/6 nephrectomy and folic acid (FA)-induced CKD

Then, we investigated the role of Insig1 in another renal fibrosis model prepared with Insig1^flox/flox and Insig1^ΔKap mice treated with 5/6 Nx for 16 weeks. In accordance with the findings from the UUO-induced CKD model, the kidneys of Insig1^ΔKap mice challenged with 5/6 Nx showed higher levels of urine albumin, SCr, and blood urea nitrogen (BUN) levels than Insig1^flox/flox mice (Fig. 3A,B). Moreover, compared to the Insig1^flox/flox mice, Insig1^ΔKap mice exhibited exacerbated collagen and FN deposition and increased expressions of the fibrotic indicators (*Fn*, *Col1a1* and *Col3a1*) after 5/6 Nx challenge (Fig. 3C,D). In addition, after 5/6 Nx challenge, Insig1 deficiency in PTCs exacerbated ER expansion and vesiculation and increased expressions of ER stress markers (Calreticulin, *Atf6* and *Trib3*), while decreasing *Slc6a13* expression and increasing *Cd74* expression (Fig. 3E–H). Furthermore, kidney functions and histologic findings in the kidneys of the FA-induced model were similar to those in the 5/6 Nx model (Fig. EV2D–F). These results revealed that Insig1 deficiency in PTCs aggravated 5/6 Nx and FA-induced renal fibrosis and ER stress.

### Insig1 deficiency in fibroblast did not worsen renal fibrosis

Because fibroblast activation plays a crucial role in renal fibrosis, we created fibroblast-specific Insig1-knockout (Insig1^ΔS100A4) mice, which had ~85% reduction in Insig1 expression in cultured primary renal interstitial fibroblasts than those of Insig1^flox/flox mice (Fig. 4A–C). In contrast to the findings with Insig1^ΔKap mice, fibroblast deficiency of Insig1 did not worsen UUO-induced renal

fibrosis, as evidenced by unchanged collagen deposition and fibrotic indicator (*Fn*, *Acta2*, *Col3a1*, and *Vim*) mRNA expressions in the kidneys (Fig. 4D,E). In addition, Insig1 deficiency in fibroblast did not worsen renal fibrosis following the 5/6 Nx challenge (Fig. 4F–H). These findings clearly illustrated the detrimental role of tubular Insig1 deficiency in renal fibrosis.

## Silencing Insig1 aggravated TGF-β1-induced fibrotic responses in vitro

To examine the direct effect of Insig1 on renal tubular cells, we transfected human renal tubular epithelial cells (HK2) with Insig1 siRNA or Ctrl siRNA (Fig. EV3A) and then treated the cells with TGF-β1 for 24 h. In accordance with our in vivo findings, silencing Insig1 worsened TGF-β1-induced fibrotic responses, as shown by increased expression of *Fn* and *Acta2*, as well as worsened ER stress compared to that of the control group (Fig. EV3B,C). We also found that silencing Insig1 boosted TGF-β1-induced expression of FN, CTGF, *Acta2*, and *Col1a1* in mouse renal tubular epithelial cells (TKPTS) (Fig. 5A–E). Therefore, we hypothesized that silencing Insig1 exacerbated TGF-β1-induced fibrotic responses in HK2 and TKPTS cells.

In contrast, overexpression of Insig1 reduced TGF-β1-induced expression of FN, CTGF, *Acta2*, and *Col1a1* compared to that of the control group (Fig. 5F–J). To evaluate whether aggravated TGF-β1-induced fibrotic responses in Insig1 silencing PTCs is related to ER stress, Insig1 silencing cells were treated with 4-Phenylbutyric acid (4-PBA), an inhibitor of ER stress, to antagonize ER stress before TGF-β1 treatment. As expected, 4-PBA markedly ameliorated TGF-β1-induced fibrotic responses in both control and Insig1 silencing PTCs (Fig. EV3D). Collectively, these findings showed that Insig1 significantly inhibited the synthesis of extracellular matrix components and ER stress caused by TGF-β1, showing that Insig1 functions as a protective factor against renal fibrogenesis.

## Aldh1a1 was a transcriptional target of Insig1 in the kidney

As Insig1 regulates lipid metabolism, we first examined whether Insig1 deletion exacerbates UUO-induced renal fibrosis mediated through its effect on lipid metabolism. We observed that UUO-induced lipogenesis (triacylglycerol (TG) accumulation and the activity of *Acc1* and *Fas*) was unaffected by Insig1 deletion

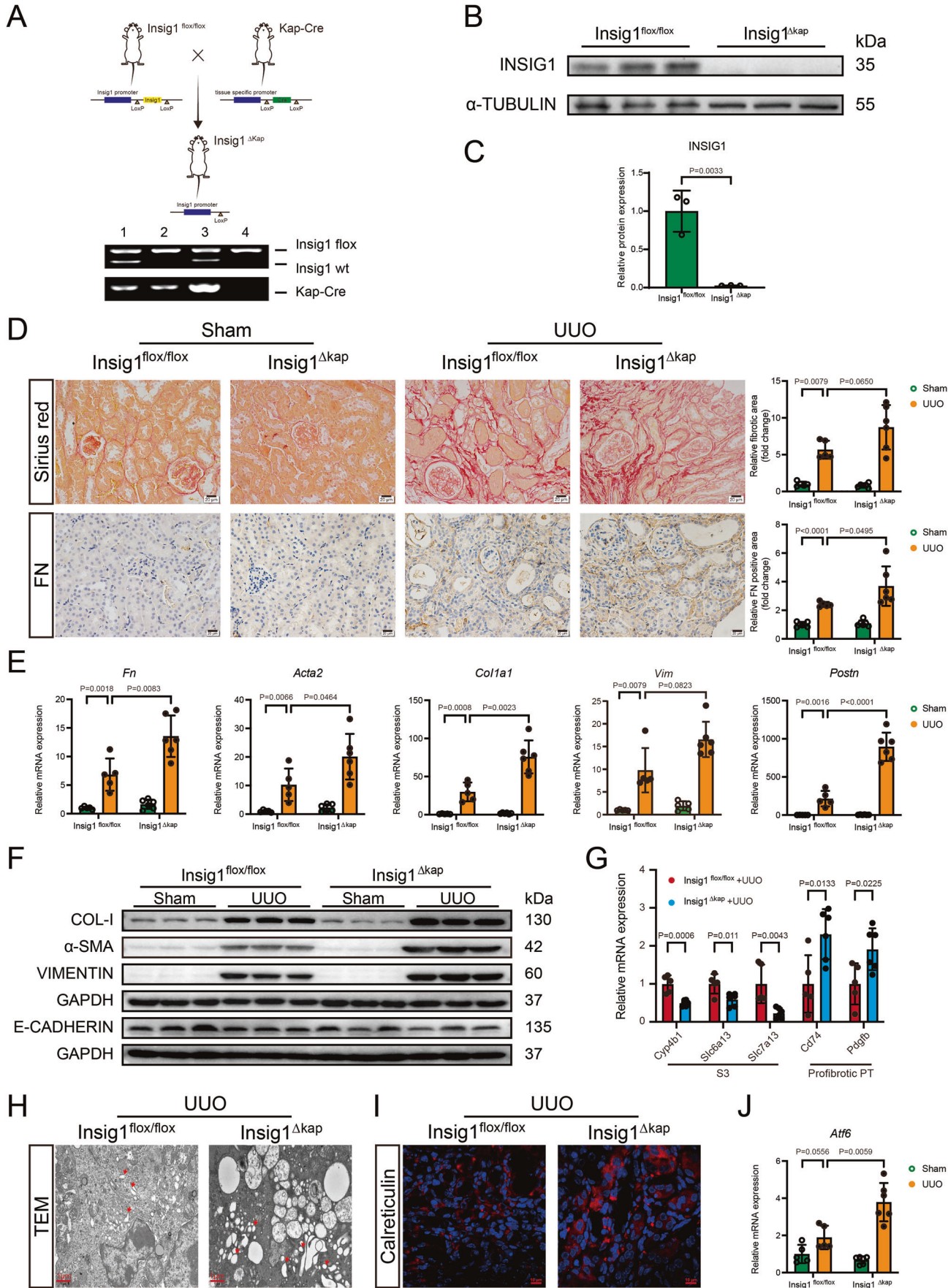

**Figure 2. Insig1 deficiency in PTCs aggravated UUO-induced CKD.**

(A) Figdraw produced a schematic diagram showing the strategy for generating Insig1$^{\Delta Kap}$ mice (#1, #3: Insig1$^{flox/wt}$, Kap-cre +; #2: Insig1$^{flox/flox}$, Kap-cre +; #4: Insig1$^{flox/flox}$, Kap-cre −). (B, C) Representative immunoblotted bands and quantification of Insig1 in isolated renal tubule tissue ($n = 3$ in each group, biological replicates). (D) Representative images and quantification of Sirius red staining and IHC (FN) staining in mouse kidneys (scale bars, 20 μm; $n = 5$ in Insig1$^{flox/flox}$ group and Insig1$^{flox/flox}$ + UUO group; $n = 6$ in Insig1$^{\Delta Kap}$ group and Insig1$^{\Delta Kap}$ + UUO group, biological replicates). (E) qPCR analysis of *Fn, Acta2, Col1a1, Postn* and *Vim* in the kidneys of Insig1$^{flox/flox}$ and Insig1$^{\Delta Kap}$ mice after UUO ($n = 5$ in Insig1$^{flox/flox}$ group and Insig1$^{flox/flox}$ + UUO group; $n = 6$ in Insig1$^{\Delta Kap}$ group and Insig1$^{\Delta Kap}$ + UUO group, biological replicates). (F) Representative immunoblot bands showing COL-I, α-SMA, VIMENTIN, and E-CADHERIN ($n = 5$ in Insig1$^{flox/flox}$ group and Insig1$^{flox/flox}$ + UUO group; $n = 6$ in Insig1$^{\Delta Kap}$ group and Insig1$^{\Delta Kap}$ + UUO group, biological replicates). (G) qPCR analysis of profibrotic PT and S3 subset markers ($n = 5$ in Insig1$^{flox/flox}$ group and Insig1$^{flox/flox}$ + UUO group; $n = 6$ in Insig1$^{\Delta Kap}$ group and Insig1$^{\Delta Kap}$ + UUO group, biological replicates). (H, I) Representative electron micrographs of ER in the PTCs (scale bars, 1 μm, red arrows denote expanded ER) and calreticulin in the mouse kidneys (scale bars, 10 μm; $n = 3$ in each group, biological replicates). (J) qPCR analysis of *Atf6* in the kidneys of Insig1$^{flox/flox}$ and Insig1$^{\Delta Kap}$ mice after UUO ($n = 5$ in Insig1$^{flox/flox}$ group and Insig1$^{flox/flox}$ + UUO group; $n = 6$ in Insig1$^{\Delta Kap}$ group and Insig1$^{\Delta Kap}$ + UUO group, biological replicates). Data information: In (C), Data are represented as mean ± SD. Student's *t* test. (D, E, G, J) Data are represented as mean ± SD. Student's *t* test or Mann–Whitney test. Source data are available online for this figure.

(Fig. EV3E,F). In addition, TOFA (an inhibitor of acetyl-CoA carboxylase-α) and C75 (an inhibitor of fatty acid) were applied to an in vitro CKD model. When TGF-β1 was present, silencing Insig1 enhanced the mRNA expression of *Fn* and *Ctgf*, and neither TOFA nor C75 reversed this trend (Fig. EV3G). These findings suggested that lipogenic activity may not be related to the profibrotic effect of Insig1 deletion in PTCs.

Then, utilizing isolated renal tubule tissue, we conducted a bulk RNA-seq analysis to examine differences in gene expression between Insig1$^{flox/flox}$ and Insig1$^{\Delta Kap}$ mouse PTCs to better understand the underlying mechanism by which Insig1 prevents renal fibrosis. Insig1$^{\Delta Kap}$ mice showed 179 differentially expressed genes, 109 downregulated genes, and 70 upregulated genes, in comparison to Insig1$^{flox/flox}$ mice (Fig. 6A). The retinol metabolism pathway was one of the most significantly altered pathways, according to KEGG pathway enrichment results (Fig. 6B). Aldh1a1, a crucial enzyme in retinol metabolism, was found to be significantly enhanced in Insig1-silenced TKPTS cells by analyzing the change in gene expression in retinol metabolism (Fig. 6C,D). Aldh1a1 was also markedly increased in the kidneys of Insig1$^{\Delta Kap}$ mice in contrast to Insig1$^{flox/flox}$ mice (Fig. 6E). In addition, the luciferase reporter gene data showed that Insig1 negatively modulated Aldh1a1 transcriptional activity (Fig. 6F,G).

According to the literature, after deletion, Insig1 was unavailable to interact with SCAP, causing the SCAP-SREBP1 complex to translocate to the Golgi apparatus and increasing the amount of SREBP1 entering the nucleus (Gong et al, 2006). Moreover, SREBP1 protein expression was markedly decreased in the cytoplasm but increased in the nucleus of Insig1-silenced TKPTS cells (Fig. 6H,I), which suggests silencing Insig1 boosted SREBP1 activation. Aldh1a1 transcriptional activity was significantly elevated when SREBP1 was overexpressed (Fig. 6J). A chromatin immunoprecipitation (ChIP) assay revealed that SREBP1 bound to the Aldh1a1 promoter region (Fig. 6K). In addition, fatostatin (a specific inhibitor of SREBP activation) markedly reversed the increase in transcriptional activity of Aldh1a1 in Insig1-silenced TKPTS cells (Fig. 6L). Although the active form of SREBP1 (1–480 bp) increased Aldh1a1 transcriptional activity, fatostatin was failed to reverse this trend (Fig. 6M). These findings demonstrated that Insig1 prevented SREBP1 protein entry into the nucleus, thereby lowering Aldh1a1 transcriptional activity.

## Inhibiting Aldh1a1 alleviated renal fibrosis in vivo and in vitro

The Aldh1a1 level was found to be significantly higher in the UUO model mice (Figs. 7A and EV4A,B), and it positively correlated with

the SCr level and negatively correlated with the eGFR (Fig. 7B). To verify the role of Aldh1a1 in CKD, we subsequently employed Aldh1a1-KO mice (Fig. EV4C,D). In contrast to wild-type (WT) mice, Aldh1a1 deletion markedly decreased collagen deposition and FN expression in the kidneys after UUO (Fig. 7C). The fibrotic index (COL-I, *Acta2, Col3a1, Postn*, and *Ctgf*) and the profibrotic PT maker (*Pdgfb*) level were also significantly lower in the Aldh1a1-KO animals after UUO, while the S3 marker (*Slc6a13, Slc7a13*, and *Cyp4b1*) levels were significantly higher (Figs. 7D–F and EV4E). Moreover, Aldh1a1 deletion significantly decreased ER dilation and ER stress makers (Calreticulin, *Atf6*, and *Perk*) (Fig. 7G–I).

In TKPTS cells, overexpression of Aldh1a1 (Fig. EV4F) worsened TGF-β1-induced fibrotic responses, as shown by increased expression of FN, *Acta2* and *Ctgf* (Fig. 7J,K). Conversely, when compared to the control group, Aldh1a1 silencing decreased ER stress and TGF-β1-induced FN and α-SMA expression in TKPTS cells (Figs. 7L and EV4G–I). In addition, modifying the protein's active site of Aldh1a1 may be able to block its profibrotic effects (Fig. 7M).

In line with these findings, pharmacological intervention using the Aldh1a1 inhibitor NCT501 reduced the fibrotic responses induced by TGF-β1 in TKPTS and HK2 cells (Fig. EV5A–D). These results together demonstrated that Aldh1a1 overexpression exacerbated renal fibrosis.

## Insig1/Aldh1a1 activation mitigated renal fibrosis by increasing NAD$^+$ levels

An untargeted metabolomics analysis was carried out to determine the mechanism of Aldh1a1 in renal fibrosis. As expected, we discovered a considerable decrease in the levels of NAD$^+$ and its precursors (nicotinamide riboside and nicotinamide ribotide) in kidneys after UUO (Appendix Fig. S1A–C), and this decline was even more pronounced following the loss of Insig1 in PTCs (Fig. 8A,B; Appendix Fig S1D,E). Consistent with the metabolomic analysis results, after UUO or 5/6 Nx challenge, the NAD$^+$ concentration was considerably lower in the kidneys of the Insig1$^{\Delta Kap}$ mice than in the kidneys of the Insig1$^{flox/flox}$ mice. In addition, compared to WT mice, Aldh1a1-KO mice showed significantly higher levels of NAD$^+$ in the kidneys after UUO (Fig. 8C). NCT501 treatment increased the NAD$^+$/NADH ratio in TKPTS cells despite Insig1 silencing or Aldh1a1 overexpression decreasing it. In addition, Aldh1a1 overexpression worsened TGF-β1-induced NAD$^+$ consumption (Fig. 8D,E). These findings suggested that the anti-fibrotic activity of the Insig1/Aldh1a1 pathway may depend on NAD$^+$ homeostasis.

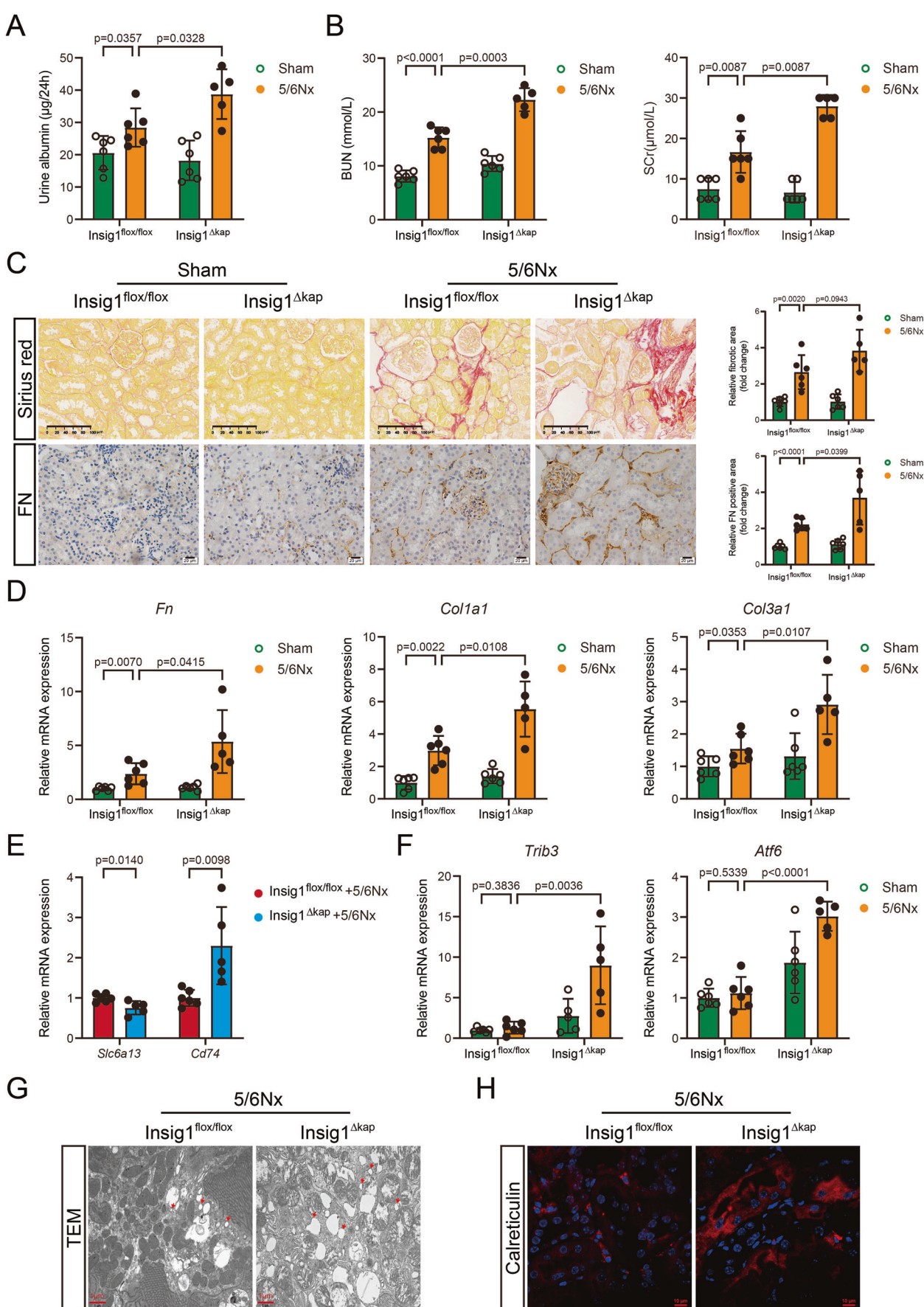

**Figure 3. Insig1 deficiency in PTCs aggravated 5/6 nephrectomy-induced CKD.**

(A) Twenty-four-hour urine albumin concentrations were measured in 5/6 Nx-treated Insig1$^{flox/flox}$ and Insig1$^{\Delta Kap}$ mice ($n = 6$ in Insig1$^{flox/flox}$ group, Insig1$^{flox/flox}$ + 5/6 Nx group and Insig1$^{\Delta Kap}$ group; $n = 5$ in Insig1$^{\Delta Kap}$ + 5/6 Nx group, biological replicates). (B) SCr and BUN concentrations were measured in 5/6 Nx-treated Insig1$^{flox/flox}$ and Insig1$^{\Delta Kap}$ mice ($n = 6$ in Insig1$^{flox/flox}$ group, Insig1$^{flox/flox}$ + 5/6 Nx group and Insig1$^{\Delta Kap}$ group; $n = 5$ in Insig1$^{\Delta Kap}$ + 5/6 Nx group, biological replicates). (C) Representative images and quantification of Sirius red staining (scale bars, 100 μm) and IHC (FN, scale bars, 20 μm) in mouse kidneys ($n = 6$ in Insig1$^{flox/flox}$ group, Insig1$^{flox/flox}$ + 5/6 Nx group and Insig1$^{\Delta Kap}$ group; $n = 5$ in Insig1$^{\Delta Kap}$ + 5/6 Nx group, biological replicates). (D) qPCR analysis of *Fn*, *Col1a1*, and *Col3a1* in the kidneys of Insig1$^{flox/flox}$ and Insig1$^{\Delta Kap}$ mice subjected to 5/6 Nx ($n = 6$ in Insig1$^{flox/flox}$ group, Insig1$^{flox/flox}$ + 5/6 Nx group and Insig1$^{\Delta Kap}$ group; $n = 5$ in Insig1$^{\Delta Kap}$ + 5/6 Nx group, biological replicates). (E) qPCR analysis of profibrotic PT subset markers (*Cd74*) and S3 subset markers (*Slc6a13*; $n = 6$ in Insig1$^{flox/flox}$ + 5/6 Nx group; $n = 5$ in Insig1$^{\Delta Kap}$ + 5/6 Nx group, biological replicates). (F) qPCR analysis of *Atf6* and *Trib3* in the kidneys of Insig1$^{flox/flox}$ and Insig1$^{\Delta Kap}$ mice subjected to 5/6 Nx ($n = 6$ in Insig1$^{flox/flox}$ group, Insig1$^{flox/flox}$ + 5/6 Nx group and Insig1$^{\Delta Kap}$ group; $n = 5$ in Insig1$^{\Delta Kap}$ + 5/6 Nx group, biological replicates). (G, H) Representative electron micrographs of ER in the PTCs (scale bars, 1 μm; red arrows denote expanded ER) and calreticulin in the mouse kidneys (scale bars, 10 μm; $n = 3$ in each group, biological replicates). Data information: In (A, C, E, F), Data are represented as mean ± SD. Student's *t* test. (B, D) Data are represented as mean ± SD. Student's *t* test or Mann–Whitney test. Source data are available online for this figure.

Recent research suggested that NAD$^+$ controlled ER membrane expansion by deribosylating the ER sensor GRP78/BIP (Wu et al, 2021). In TKPTS cells, NCT501 treatment or Insig1 overexpression significantly increased ADP-ribosylated BIP levels (Fig. 8F,G). Insig1 silencing also resulted in an expansion of the ER membrane and reduced IRE1α and BIP binding. NAD$^+$ treatment or Aldh1a1 silencing, however, increased BIP and IRE1α binding (Fig. 8H–J). Together, Insig1 overexpression or Aldh1a1 silencing restored the ADP-ribosylation of BIP, enhanced the binding of IRE1α and BIP, and maintained ER homeostasis by increasing NAD$^+$ levels.

## Identification of a novel human Aldh1a1 inhibitor that alleviated UUO-induced CKD in an Aldh1a1-dependent manner

The TargetMol-Approved Drug Screening Library (a total of 1600 compounds) and TargetMol-Anti Cancer Compound Library (a total of 3145 compounds) were used in our study for virtual screening (Fig. 9A). On the basis of the virtual screening results, nineteen compounds were chosen for additional examination by in vitro enzymatic activity assays. The activity of the human recombinant Aldh1a1 (hAldh1a1) protein was profoundly inhibited by nicardipine (13-F9), and its docking score was −9.4128 (Fig. 9B,C). Then the BIAcore system was used to directly evaluate the binding of nicardipine and Aldh1a1. The equilibrium dissociation constant (KD) value of Aldh1a1 with nicardipine was 1.98E-06 M, indicating strong binding between nicardipine and Aldh1a1, while the KD value of Aldh1a1 with NCT501 was 3.31E-05 M (Fig. 9D; Appendix Fig S2A). In vivo, nicardipine treatment decreased Aldh1a1 protein expression in the kidney but exerted no effect on Aldh1a1 mRNA expression (Fig. 9E; Appendix Fig S2B). Consistent with in vivo results, nicardipine lowered Aldh1a1 protein expression in TKPTS cells but exerted no effect on Aldh1a1 mRNA expression or transcriptional activity (Fig. 9F; Appendix Fig S2C,D). In addition, the protease inhibitor MG132 profoundly reversed the nicardipine-induced reduction in Aldh1a1 protein expression (Appendix Fig. S2E). Thus, a proteasomal degradation mechanism may be the mechanism utilized by nicardipine to inhibit Aldh1a1 expression. However, other ALDHs (ALDH2, ALDH1A2, ALDH1B1)'s protein expressions were not decreased by nicardipine (Appendix Fig. S2F,G). Thus, nicardipine may function as a novel Aldh1a1 inhibitor.

For further analysis of the effects of nicardipine on UUO-induced CKD, mice were given nicardipine (1 mg/kg/d intragastrically (i.g.)) for 2 days before UUO surgery. Nicardipine administration decreased renal fibrosis in the mice after UUO, which was consistent with the outcomes observed in NCT501-treated cells. Following nicardipine treatment, collagen deposition, and FN expression were reduced in the mice after UUO (Fig. 9G), and profibrotic marker expression (FN, CTGF, *Col1a1*, and *Col3a1*), ER stress index (*Atf6*, *Trib3*), and ER expansion, were decreased in the kidneys compared to UUO-only exposed animals (Fig. 9H–K). Similarly, nicardipine markedly reduced the fibrotic reactions caused by TGF-β1 in TKPTS cells (Fig. EV5E–G). Mechanistically, nicardipine greatly elevated the NAD$^+$ concentration and the NAD$^+$/NADH ratio, which had been decreased by TGF-β1, significantly boosting ADP-ribosylated BIP levels and increasing IRE1α and BIP binding, which prevented ER expansion (Fig. 9L–N).

To determine whether the protective effect of nicardipine on UUO-induced CKD involved an Aldh1a1-dependent mechanism, we administered nicardipine to WT and Aldh1a1-KO mice for 14 days. As expected, the protective effects of nicardipine on UUO-induced CKD were abolished in Aldh1a1-KO mice (Fig. 9O–Q), showing that Aldh1a1 is involved in the effect of nicardipine on UUO-induced CKD. In addition, we compared the therapeutic effect of nicardipine and NCT501 in the UUO-induced CKD model. Compared to NCT501, nicardipine treatment significantly reduced collagen deposition and the fibrotic indicators expression in the kidneys following UUO surgery (Fig. EV5H,I). According to these findings, nicardipine might be a new highly targeted Aldh1a1 inhibitor.

## Discussion

Based on the results of genetic, in vivo, in vitro, and pharmacological investigations, this study revealed that PTC-specific Insig1 plays a significant role in the progression of renal fibrosis in CKD. Here, we found an apparent reduction in Insig1 in the kidney under CKD conditions. Furthermore, we described the pathogenic impact of PTC-specific Insig1 deficiency on renal fibrosis in vivo and in vitro, as opposed to fibroblast-specific Insig1 deficiency. Overexpression of Insig1 in PTCs potently inhibited TGF-β1-induced fibrotic responses. All these in vivo and in vitro experimental data strongly suggested that tubular Insig1 deficiency aggravated renal fibrosis in CKD.

Re-analyzing the PTs subsets in previously uploaded scRNA-seq data (Doke et al, 2022), we discovered that Insig1 was a branch-specific gene that was crucial for controlling how the S1 subset transitioned

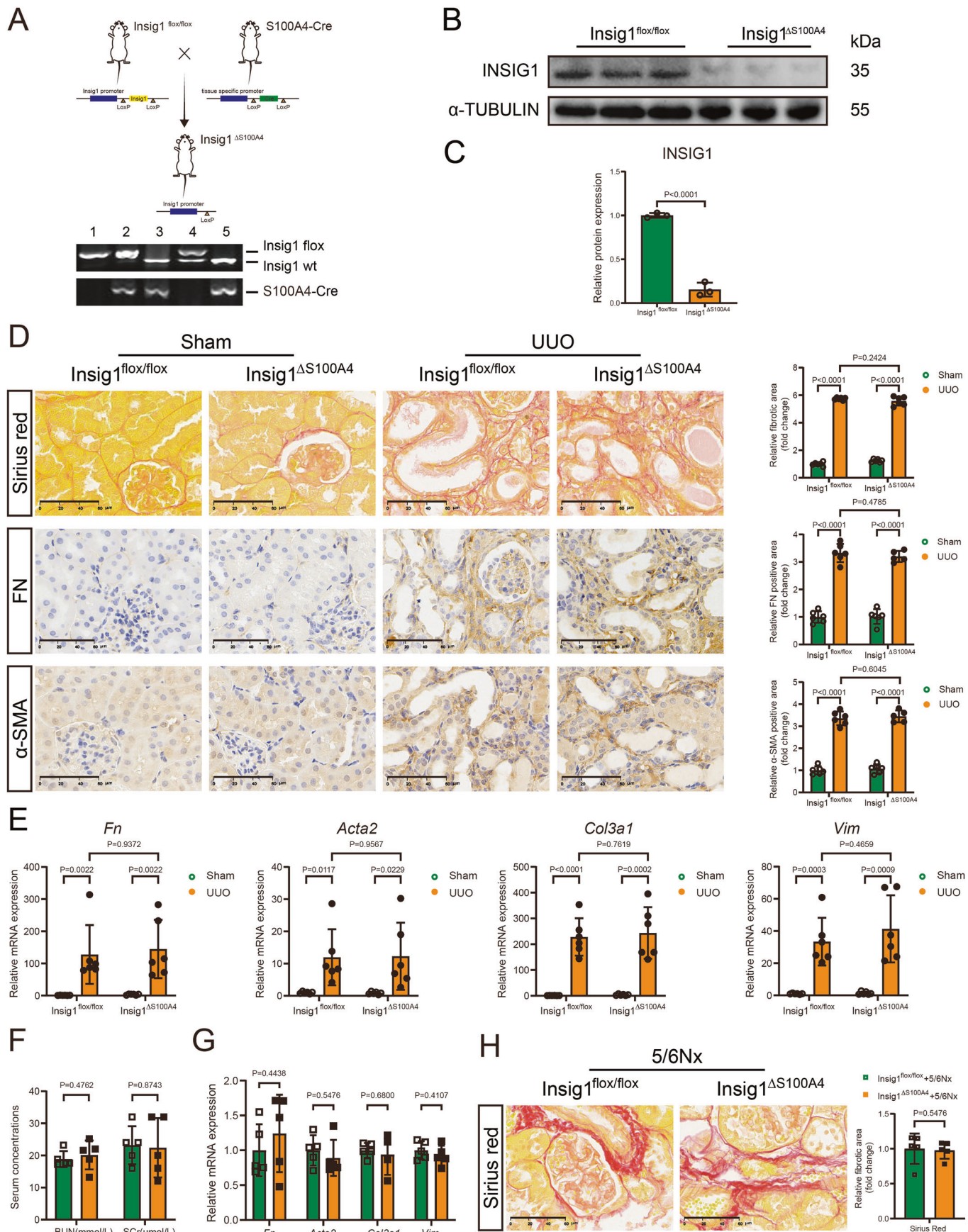

◄

**Figure 4. Insig1 deficiency in fibroblast did not worsen renal fibrosis.**

(A) Figdraw produced a schematic diagram showing the strategy for generating Insig1$^{\Delta S100A4}$ mice (#1: Insig1$^{flox/flox}$, Kap-cre +; #2: Insig1$^{flox/flox}$, Kap-cre -; #3, #5: Insig1$^{wt/wt}$, Kap-cre +; #4: Insig1$^{flox/wt}$, Kap-cre −). (B, C) Representative immunoblotted bands and quantification of Insig1 in cultured primary renal fibroblasts ($n = 3$, biological replicates). (D) Representative images and quantification of Sirius red staining and IHC (FN, α-SMA) staining in mouse kidneys (scale bars, 60 μm; $n = 5$–6 in each group, biological replicates). (E) qPCR analysis of *Fn*, *Acta2*, *Col3a1*, and *Vim* in the kidneys of Insig1$^{flox/flox}$ and Insig1$^{\Delta S100A4}$ mice after UUO ($n = 6$ in each group, biological replicates). (F) SCr and BUN concentrations were measured in 5/6 Nx-treated Insig1$^{flox/flox}$ and Insig1$^{\Delta S100A4}$ mice ($n = 5$ in each group, biological replicates). (G) qPCR analysis of *Fn*, *Acta2*, *Col3a1*, and *Vim* in the kidneys of Insig1$^{flox/flox}$ and Insig1$^{\Delta S100A4}$ mice after 5/6 Nx ($n = 5$ in each group, biological replicates). (H) Representative images and quantification of Sirius red staining in mouse kidneys (scale bars, 60 μm; $n = 5$ in each group, biological replicates). Data information: (C, D) Data are represented as mean ± SD. Student's *t* test. (E–H) Data are represented as mean ± SD. Student's *t* test or Mann–Whitney test. Source data are available online for this figure.

into the S3 and profibrotic PT subsets. Furthermore, by using hdWGCNA to cluster branch-dependent genes, we discovered that Insig1 was in the blue gene module. Notably, our experimental results were in line with the eigengene expression changes in PT subsets within the blue gene module. These results suggested that higher Insig1 expression could inhibit the S1 subset's branch into the profibrotic PT subset and encourage it to branch into the S3 subset.

As SREBP1 is a well-characterized lipogenic transcription factor (Lee et al, 2022), Insig1 controls lipogenesis and cholesterol metabolism by controlling SREBP1 into the nucleus (Gong et al, 2006). We investigated whether the profibrotic effect of tubular Insig1 deficiency may be mediated by lipogenic activity. However, we found that the lipogenesis caused by UUO was unaffected by tubular Insig1 deficiency. In line with this finding, when Insig1 was knocked down in TKPTS cells, TOFA and C75 were unable to block the profibrotic effect of TGF-β1. Thus, the profibrotic effect of Insig1 deletion in PTCs did not appear to be linked to lipogenic activity.

In our research, we discovered for the first time that Insig1 may control the transcription of Aldh1a1 via the action of SREBP1. When Insig1 was deleted, SREBP1 nuclear translocation was enhanced, and Aldh1a1 transcription was subsequently activated. The transcriptional inhibition of Aldh1a1 by Insig1 was reversed when SREBP1 was prevented from moving to the Golgi apparatus from the ER by fatostatin treatment. Furthermore, Aldh1a1's profibrotic effect was reversed by altering the enzyme's active site to minimize NAD$^+$ consumption. Together, these data suggested that Insig1 maintained NAD$^+$ homeostasis through the transcriptional repression of Aldh1a1.

NAD$^+$ preservation is increasingly being recognized as a way of delaying the effects of ageing on health and thus potentially extend lifetime (Chowdhry et al, 2019; Covarrubias et al, 2021; Poyan Mehr et al, 2018). The kidney is a crucial organ in de novo NAD$^+$ biosynthesis (Spath et al, 2023). In AKI and CKD, de novo NAD$^+$ biosynthesis is impaired, and substantial decreases in the levels of NAD$^+$ can reduce energy generation (Poyan Mehr et al, 2018). More recently, NAD$^+$ augmentation through supplementation with nicotinamide mononucleotide (NMN) or nicotinamide (NAM), an NAD$^+$ precursor, has been proposed as for the prevention of numerous kidney illnesses, including AKI, CKD, the AKI-to-CKD transition, and diabetic nephropathy (Clark et al, 2022; Guan et al, 2017; Jia et al, 2021; Kumakura et al, 2021; Ralto et al, 2020; Yasuda et al, 2021; Zheng et al, 2019). Our findings demonstrated that NAD$^+$ levels were lower in tubular Insig1 deficiency-caused CKD mouse models, resulting in more severe ER stress and kidney fibrosis.

The function of the ER stress sensor GRP78/BIP depends on NAD$^+$-dependent ADP-ribosylation. By deribosylating BIP and forcing it to separate from IRE1α, NAD$^+$ deficiency resulted in ER

expansion (Wu et al, 2021). Excessive ER stress results in the persistent activation of the unfolded protein response, which is an underlying cause of PTC death and fibrosis. In addition, de novo NAD$^+$ biosynthesis in the kidney may be impaired by the ER stress response (Bignon et al, 2022), which suggested interplay between NAD$^+$ and ER stress. Here, we discovered that therapy with NCT501 or overexpression of Insig1 in TKPTS cells could restore BIP ADP-ribosylation. In addition, NAD$^+$ treatment or Insig1 overexpression increased the binding of IRE1α and BIP, but Aldh1a1 overexpression reversed this trend, suggesting that decreased Insig1 in PTCs might boost Aldh1a1 transcription activity, which in turn resulted in excessive NAD$^+$ consumption and ER expansion. The communication between the ER and mitochondria is important to a cell (Csordas et al, 2018). Lack of mitochondrial aspartate attenuated NAD$^+$ regeneration and led to BIP ADP-deribosylation (Wu et al, 2021). As a result, further exploration into how changes in CKD-related mitochondrial function are caused by Insig1 deletion in PTCs is warranted.

In this study, we first established that nicardipine was a selective Aldh1a1 antagonist. Nicardipine strongly binds Aldh1a1, increasing protein degradation and thereby decreasing Aldh1a1 activity. In addition, nicardipine exerted few harmful effects on mice (Appendix Fig. S3A–D). Nicardipine treatment was found to be more therapeutically effective than NCT501, and this protective effect was eliminated in Aldh1a1-KO mice, suggesting the involvement of Aldh1a1 in the activity of nicardipine against renal fibrosis, but we could not rule out a potential off-target effect. Mechanistically, nicardipine restored BIP ADP-ribosylation and increased the binding of IRE1α and BIP, which resulted in an increase in NAD$^+$ level and the maintenance of ER homeostasis. Furthermore, Insig1 expression was low, and Aldh1a1 expression was high in CKD patients with hypertension (Appendix Fig. S3E,F). Nicardipine is effective in the treatment of blood pressure, but here, we found that it effectively attenuated CKD, expanding its clinical indications to the treatment of CKD patients with hypertension.

A limitation of our study is that the roles of tubular Insig1 and nicardipine in ameliorating CKD were observed mainly in models of mouse and PTCs but not in humans. Although we detected the downregulation of Insig1 in patients with CKD, the effectiveness and importance of Insig1 and nicardipine in protecting against human CKD remain to be further explored in the future.

In conclusion, activation of tubular Insig1 markedly attenuated CKD, possibly by maintaining NAD$^+$ homeostasis and controlling ER expansion via the transcriptional repression of Aldh1a1 in PTCs. These findings from the present study not only increase our understanding of the pathogenesis of CKD but also highlight the therapeutic potential of activating the Insig1/Aldh1a1 pathway in

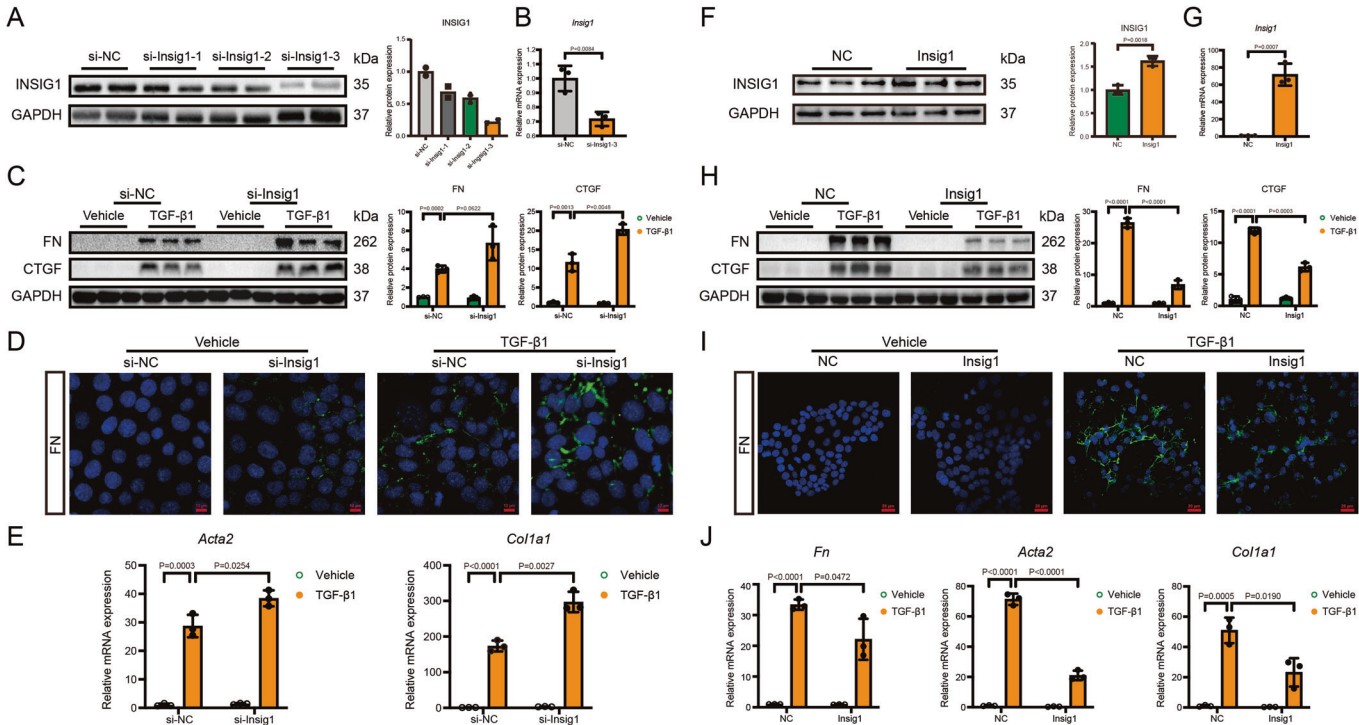

**Figure 5. Silencing Insig1 aggravated TGF-β1-induced fibrotic response in TKPTS cells.**

(A) Representative immunoblotted bands and quantification of INSIG1 in Insig1 silencing TKPTS cells (si-NC: Negative control siRNA; $n = 2$ in each group, biological replicates). (B) qPCR analysis of *Insig1* in Insig1 silencing TKPTS cells (si-Insig1-3 was selected for additional experimentation, $n = 3$ in each group, biological replicates). (C) Representative immunoblotted bands and quantification of FN and CTGF in Insig1 silencing TKPTS cells stimulated by TGF-β1 ($n = 3$ in each group, biological replicates). (D) Representative images of FN staining in Insig1 silencing TKPTS cells stimulated by TGF-β1 (scale bars, 10 μm; $n = 3$ in each group, biological replicates). (E) qPCR analysis of *Acta2* and *Col1a1* in Insig1 silencing TKPTS cells stimulated by TGF-β1 ($n = 3$ in each group, biological replicates). (F) Representative immunoblotted bands and quantification of INSIG1 in Insig1 overexpressed TKPTS cells (NC: Negative control; $n = 3$ in each group, biological replicates). (G) qPCR analysis of *Insig1* in Insig1 overexpressed TKPTS cells ($n = 3$ in each group, biological replicates). (H) Representative immunoblotted bands and quantification of FN and CTGF in Insig1 overexpressed TKPTS cells stimulated by TGF-β1 ($n = 3$ in each group, biological replicates). (I) Representative images of FN staining in Insig1 overexpressed TKPTS cells stimulated by TGF-β1 (scale bars, 20 μm; $n = 3$ in each group, biological replicates). (J) qPCR analysis of *Fn*, *Acta2*, and *Col1a1* in Insig1 overexpressed TKPTS cells stimulated by TGF-β1 ($n = 3$ in each group, biological replicates). Data information: In (B, C, E–H, J), Data are represented as mean ± SD. Student's *t* test. Source data are available online for this figure.

PTCs. Determination of the efficacy and safety of selective Aldh1a1 antagonists and future clinical trials for patients with CKD might reveal a viable approach to CKD therapy.

# Methods

### Ethical approval

The Research Ethics Board at Children's Hospital of Nanjing Medical University authorized all experiments using human tissue (202008089-1). Written informed consent was provided by each patient, and the experiments conformed to the principles set out in the WMA Declaration of Helsinki and the Department of Health and Human Services Belmont Report. All animal experiments were approved by the Institutional Animal Care and Use Committee of Nanjing Medical University (2007001). For treatment studies, the experiment or control groups were assigned at random. If possible, researchers were blinded to the treatment group; however, blinding was not possible for mouse groups because of institutional requirements for labeling Insig1$^{\Delta Kap}$ or Insig1$^{\Delta S100A4}$ or Aldh1a1$^{-/-}$ mice.

### Human kidney biopsy samples

The tissue samples of 15 diagnosed patients were provided by the Department of Nephrology, Children's Hospital of Nanjing Medical University. Normal renal tissues were collected from four patients without proteinuria who underwent a partial nephrectomy of a benign renal cell cancer. Clinical data, such as age, sex, and causes of disease, were recorded (Table EV1).

### Mice

The Animal Core Facility of Nanjing Medical University (Nanjing, China) provided male C57BL/6 mice (8–10 weeks old, 22 ± 3 g). All mice in the study were housed under controlled conditions (temperature 20–26 °C, humidity 40–70%) and a 12-h light–dark cycle.

### Generation of renal proximal tubular or fibroblast Insig1 conditional knockout mouse strains

Insig1$^{flox/wt}$ mice (T020103, C57BL/6J) were generated using CRISPR/Cas9 genome engineering by GemPharmatech (Nanjing, Jiangsu,

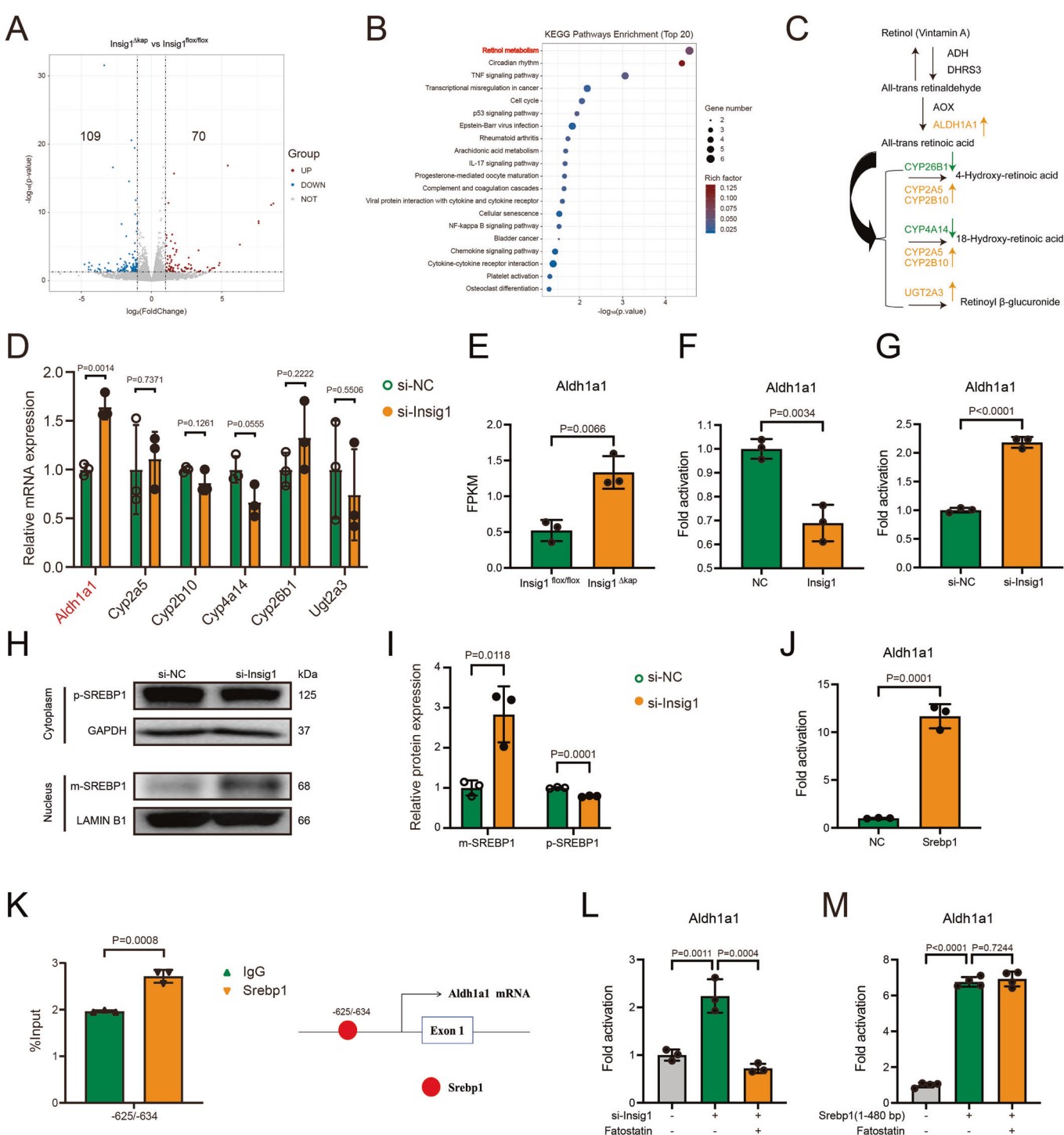

China). After that, Insig1^flox/wt self-crossing produced Insig1^flox/flox mice. Kap-Cre (The Jackson Laboratory, stock number: 008781) and S100A4-Cre mice (The Jackson Laboratory, stock number: 030644) was bred with Insig1^flox/flox mice to create F1. In order to create renal proximal tubular and fibroblast-specific conditional Insig1-knockout mice (Insig1^ΔKap and Insig1^ΔS100A4 mice), F1 was then bred with the opposite sex of Insig1^flox/flox. The control group consisted of Insig1^flox/flox mice.

## Generation of Aldh1a1-KO mouse strain

Aldh1a1-KO mice (S-KO-00984, C57BL/6J) were purchased from Cyagen (Suzhou, China). By freezing sperm resuscitation, heterozygous mice of the F0 (Aldh1a1^+/−) generation were produced. After multiple generations of hybridization, enough homozygous (Aldh1a1^−/−) mice were obtained for the experiment.

**Figure 6. Aldh1a1 was a transcriptional target of Insig1 in the kidney.**

(A) Volcano plot analysis of DEGs in the PTCs of Insig1$^{flox/flox}$ and Insig1$^{\Delta Kap}$ mice, as determined by RNA-Seq analysis. (B) KEGG analysis of DEGs between Insig1$^{flox/flox}$ and Insig1$^{\Delta Kap}$ group was examined and characterized. (C) Changes in the retinol metabolism pathway's enzyme expression. (D) qPCR analysis of *Aldh1a1, Cyp2a5, Cyp2b10, Cyp4a14, Cyp26b1, Ugt2a3* in Insig1-silenced TKPTS cells ($n = 3$ in each group, biological replicates). (E) Relative change in Aldh1a1 level in the PTCs of Insig1$^{flox/flox}$ and Insig1$^{\Delta Kap}$ mice ($n = 3$ in each group, biological replicates). (F, G) Luciferase assay of Aldh1a1 transcriptional activity in Insig1-overexpressing or Insig1-silenced TKPTS cells ($n = 3$ in each group, biological replicates). (H, I) Representative immunoblotted bands and quantification of SREBP1 in the nucleus and cytoplasm in Insig1-silenced TKPTS cells (p: precursor; m: mature; $n = 3$ in each group, biological replicates). (J) Luciferase assay of Aldh1a1 transcriptional activity in Srebp1-overexpressing TKPTS cells ($n = 3$ in each group, biological replicates). (K) ChIP analysis of Srebp1 binding to the Aldh1a1 gene promoter region in HEK293T cells ($n = 3$ in each group, biological replicates). (L, M) Luciferase assay of Aldh1a1 transcriptional activity in Insig1-silenced or Srebp1 (1–480 bp)-transfected TKPTS cells treated with or without fatostatin ($n = 3$–4 in each group, biological replicates). Data information: In (E–G, I–K), Data are represented as mean ± SD. Student's *t* test. (D) Data are represented as mean ± SD. Student's *t* test or Mann–Whitney test. (L, M) Data are represented as mean ± SD. One-way ANOVA. (A) Wald significance tests. (B) Fisher's exact test. Source data are available online for this figure.

## Unilateral ureteral obstruction (UUO)-induced fibrosis model

Mice were anesthetized, and a lateral incision was made on the back. After being exposed, the left ureter was permanently ligated by being secured at two locations with a silk suture. As sham controls, similar procedures were performed on sham-operated animals without ureteric ligation. All left renal tissues were gathered for additional examination following a 14-day modeling period. For the drug treatment experiments, 8-week-old male C57BL/6 J mice were put into three groups at random: control ($n = 6$) group; UUO-treated ($n = 6$) group; and UUO plus drug-treated ($n = 6$) group which were pretreated with NCT501 (HY-18768, MCE, 13 mg/kg/d, i.g.) or nicardipine (HY-12515A, MCE, 1 mg/kg/d, i.g. (based on its usual dosing in patients treated with nicardipine)) for 2 days, followed by UUO surgery. NCT501 or nicardipine was continuously administered for the next 14 days. The mice were sacrificed with sodium pentobarbital euthanasia (30 mg/kg, intraperitoneally (i.p.), Sigma, St Louis, MO, USA), and blood and kidney tissue were immediately collected for further study.

## 5/6 nephrectomy (5/6 Nx)-induced fibrosis model

After a midline laparotomy exposed the right kidney, the right renal artery, vein, and ureter were strangulated with silk before the kidney was completely removed. A Bovie was used to remove the poles of the left kidney. Similar procedures were performed on sham mice, however the devices were only used to touch the kidneys. The animals were injected with pentobarbital sodium (30 mg/kg, i.p.) 16 weeks after receiving 5/6 Nx.

## Folic acid-induced fibrosis model

Mice were treated with a single i.p. injection of 250 mg/kg folic acid (HY-16637, MCE) dissolved in a 0.3 M sodium bicarbonate solution. Then, mice were killed at 6 day following administration.

## Histological analysis and immunohistochemistry

The kidney sections were fixed using 4% paraformaldehyde (PFA) for 24 h, dehydrated, and eventually, embedded in paraffin. Sirius red staining (RS1220, G-CLONE, Beijing, China), Masson trichrome staining (G1346, Solarbio, Beijing, China), and Periodic Acid Schiff (PAS, G1008, Servicebio, Wuhan, China) staining were all used to stain the sections (4 or 6 μm). The stained slices were then examined using optical microscopy (Olympus BX51, Tokyo, Japan, ×400). For immunohistochemistry, the sections were deparaffinized and rehydrated using a succession of xylene and graded ethanol, then immersed in an enhanced citrate antigen retrieval solution (P0083, Beyotime, Shanghai, China), and boiled for 20 min. Following a 20-min incubation with 3% hydrogen peroxide, the resulting sections were blocked with confining liquid for 1 h at room temperature. The sections were subsequently incubated with FN (Abcam, ab2413, 1:100) for an additional night at 4 °C. The sections were examined using a DAB kit (PV-9000, ZSGB-BIO, Beijing, China) to ascertain the location of peroxidase conjugates following a 20 min incubation in a reaction intensifier and an additional 20 min incubation with horseradish peroxidase-conjugated anti-rabbit. The samples were then counterstained with hematoxylin, dehydrated, treated with xylene, and finally mounted. Slides were examined under a microscope, and the signals were analyzed using analysis tools in Image-Pro Plus software (Media Cybernetics, Silver Spring, USA).

## Immunofluorescence (IF) staining

IF staining was performed as previously described (Yu et al, 2020). Cells were incubated with an FN antibody (Abcam, ab2413, 1:100), LTL (Vector Laboratories, FL-1321-2, 1:100), PDGFB (Santa Cruz, sc-365805, 1:200) or Calreticulin antibody (CST, 12238, 1:300), which was diluted in 3% BSA, overnight at 4 °C. Subsequently, the cells were incubated with Goat anti-mouse IgG secondary antibody, Alexa Fluor 594 (Abcam, ab150116, 1:200), Alexa Fluor 488 (Abcam, ab150113, 1:200) and Goat anti-rabbit IgG secondary antibody, Alexa Fluor 488 (Invitrogen, A32731TR, 1:200), Alexa Fluor 594 (Invitrogen, A11012, 1:200) for 60 min at room temperature and then stained with the nuclear-specific stain DAPI (Beyotime, C1005) for 5 min at room temperature. Finally, the cells were visualized using laser scanning confocal microscope (Carl Zeiss LSM710, Oberkochen, Germany).

## Primary renal epithelial cell isolation

Primary proximal tubular cells were obtained from Insig1$^{\Delta Kap}$ and Insig1$^{flox/flox}$ mice. Briefly, male mice (6–8 weeks old) were euthanized, and the cortex of the kidney surface were immediately collected and placed in cold HBSS with 1% penicillin and streptomycin. Then, the kidney cortex were minced into pieces of ~1 mm$^3$ and digested in 5 ml HBSS containing 2 mg/ml collagenase I for 30 min at 37 °C. The supernatants were passed through a 100

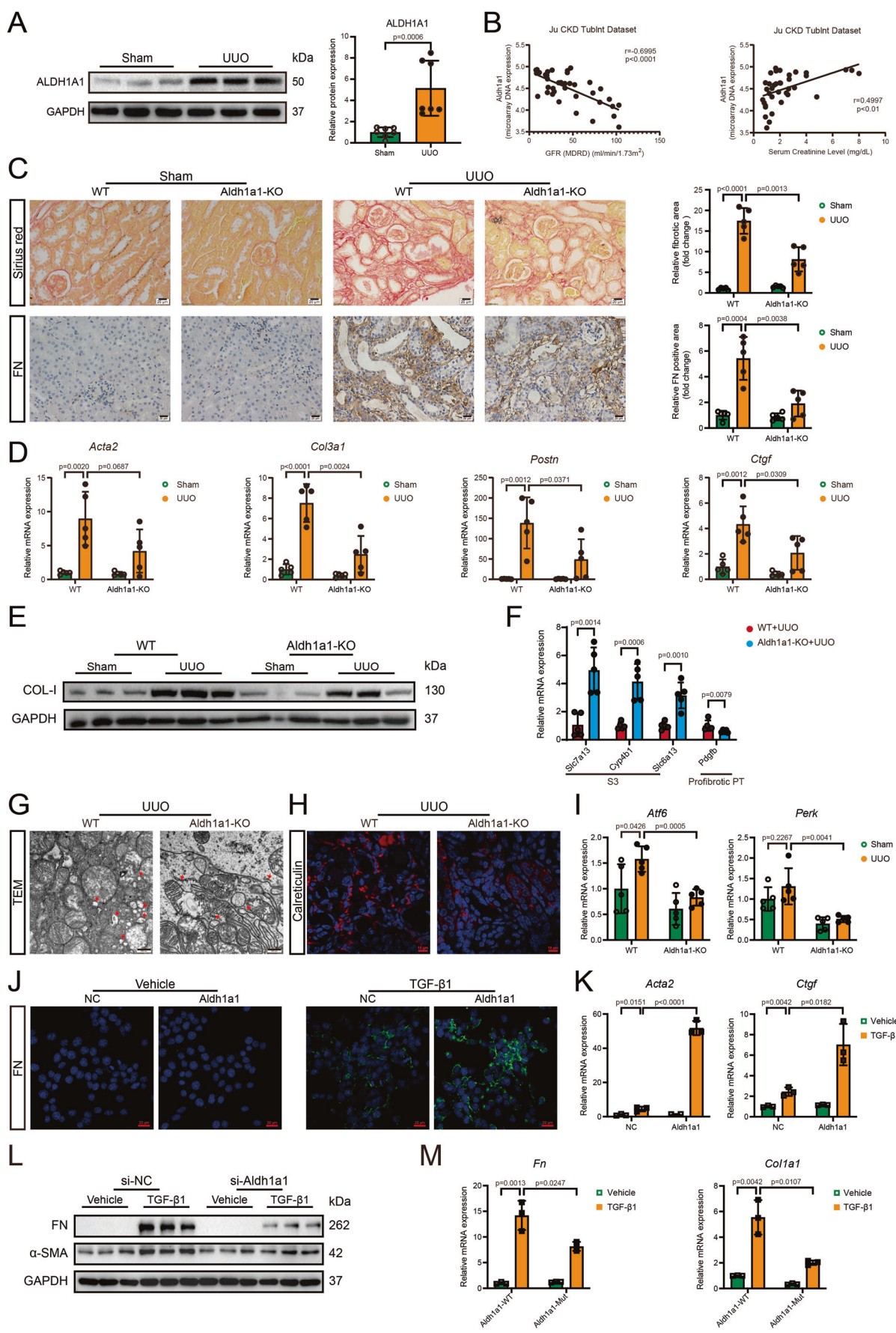

**Figure 7. Inhibiting Aldh1a1 alleviated renal fibrosis in vivo and in vitro.**

(A) Representative immunoblotted bands and quantification of ALDH1A1 in the kidneys of UUO model ($n = 7$). (B) Pearson's correlation of the Aldh1a1 level with the GFR and SCr level (GFR, $n = 38$; SCr, $n = 38$). (C) Representative images and quantification of Sirius red staining and IHC (FN) in kidneys of mice (scale bars, 20 μm; $n = 5$ in each group, biological replicates). (D) qPCR analysis of *Acta2*, *Col3a1*, *Postn*, and *Ctgf* in the kidneys of WT and Aldh1a1-KO mice after UUO ($n = 5$ in each group, biological replicates). (E) Representative immunoblotted bands of COL-I ($n = 5$ in each group, biological replicates). (F) qPCR analysis of profibrotic PT and S3 subset markers ($n = 5$ in each group, biological replicates). (G, H) Representative electron micrographs of ER in the PTCs (scale bars, 500 nm; red arrows denote expanded ER; $n = 2$ in each group, biological replicates) and calreticulin in the mouse kidneys (scale bars, 10 μm; $n = 3$ in each group, biological replicates). (I) qPCR analysis of *Atf6* and *Perk* in the kidneys of WT and Aldh1a1-KO mice after UUO ($n = 5$ in each group, biological replicates). (J, K) Representative images of FN staining (scale bars, 20 μm) and qPCR analysis of *Acta2* and *Ctgf* in Aldh1a1-overexpressing TKPTS cells stimulated by TGF-β1 ($n = 3$ in each group, biological replicates). (L) Representative immunoblotted bands of FN and α-SMA in Aldh1a1-silenced TKPTS cells stimulated by TGF-β1 ($n = 3$ in each group, biological replicates). (M) qPCR analysis of *Fn* and *Col1a1* in Aldh1a1-mutated TKPTS cells stimulated by TGF-β1 ($n = 3$ in each group, biological replicates). Data information: In (C, D, I, K, M), Data are represented as mean ± SD. Student's *t* test. (A, F) Data are represented as mean ± SD. Student's *t* test or Mann–Whitney test. Source data are available online for this figure.

mesh sieves, and then centrifuged for 10 min at $900 \times g$ and 4 °C. The primary cells were cultured with DMEM/F12 supplemented with 10% FBS, 20 ng/mL EGF (Sigma, St. Louis, MO), and maintained at 37 °C and 5% $CO_2$ in a humidified incubator.

## Primary renal fibroblast isolation

Primary renal fibroblasts were obtained from Insig1$^{\Delta S100A4}$ and Insig1$^{flox/flox}$ mice. Briefly, male mice (6–8 weeks old) were euthanized, and the cortex of the kidney surface were cut off under aseptic conditions, and then the medullary tissues were cut into ~1 mm$^3$ after washing with cold HBSS. Then, 0.1% collagenase III was added and digested at 37 °C for 10 min. The supernatant was passed through 100 mesh and 150 mesh sieves in turn, and the interstitial fragments rich in renal tubules on 150 mesh sieves was collected in DMEM medium. The primary cells were cultured with DMEM supplemented with 20% FBS, 2 μg/mL α-FGF (Sigma, St. Louis, MO) and maintained at 37 °C and 5% $CO_2$ in a humidified incubator.

## Biacore assays

We investigated the nicardipine binding kinetics to human Aldh1a1 using a Biacore T200 device (GE Healthcare Co., Stockholm, Sweden). The recombinant human Aldh1a1 protein (Novoprotein, CF44) was immobilized on the CM5 sensor chip (GE Healthcare) in accordance with standard procedures using the amine coupling method. Nicardipine was prepared at final concentrations of 10, 5, 2.5, 1.25, 0.625, and 0 (blank) μM by dilution into the running buffer. The equilibrium dissociation constant (Kd), which assesses the strength of the interaction between nicorandil and the hAldh1a1 proteins, was calculated using the Biacore T200 Evaluation program.

## Re-analysis of single-cell RNA sequencing dataset GSE182256

### Data collection

The single-cell RNA sequencing dataset GSE182256 was obtained from GEO dataset. It contains six sham kidneys and two UUO kidneys, the mice model was obtained by ligation of the left ureter for 7 days. This dataset mainly used to decode the evolution process of fibrotic PTs. The public bulk RNA-seq GSE118341 was used to calculate the differentially expressed genes (DEGs), we used three normal samples and four UUO samples on day 7.

### Data process and cell type identification

We used R package Seurat 4.0 to process the single-cell RNA sequencing dataset. The cells containing more 50% mitochondrial genes, or feature genes less than 200 or more 3000 were set as low-quality cells and removed in subsequent analysis. Then, the normalized and scaled top 2000 highly variable genes were used to obtain the principal component, and harmony integration was applied to reduce the batch effect. Finally, the cells were visualized by UMAP and cell types were annotated by canonical markers.

### Trajectory inference and branch-dependent genes analysis

The fibrosis pseudo-time trajectory of PT in UUO model was inferred by R package monocle2. The genes used to order cells were selected from the DEGs of PT subsets, the discriminative dimensionality reduction with trees (DDRTree) method was used to order cells. The branch containing most precursor PT or normal cells was set as the root. The branched expression analysis modeling (BEAM) method was used to get the PT subsets transition-related genes. Then the gene modules were clustered by the hdWGCNA based on the gene expression profiles of different cell types. The biological function of interest gene sets was enriched by gene ontology analysis.

## Screening novel human Aldh1a1 inhibitors

### Molecular docking

The Chinese Academy of Sciences Center for Excellence in Molecular Cell Sciences' compound platform conducted virtual screening. For virtual screening, the TargetMol-Approved Drug Screening Library and the TargetMol-Anti cancer compound Library (http://sjzx.sibcb.ac.cn/Cn/Index/listView/catid/58.html) were adopted. As a receptor structure, hAldh1a1's crystal structure (PDB: 4X4L) was used. Using Glide 7.5 (Schrödinger, New York, NY, USA) in the SP (standard precision) mode, docking was carried out, and compounds with a score better than −8.5 were chosen as potential hits.

### Enzymatic assays

19 substances were chosen for additional investigation utilizing the enzymatic assays based on the outcomes of the virtual screening. The following methods were used to assess a compound's ability to inhibit hAldh1a1: 100–200 nM enzyme, 500 μM NAD$^+$ (HY-B0445, MCE), 10 μM compounds, and 400 μM propionaldehyde in 20 mM Tris-HCl, pH = 7.5 at 25 °C. Fluorescence intensity change was measured at 405 nm over a 15–30 min reaction period.

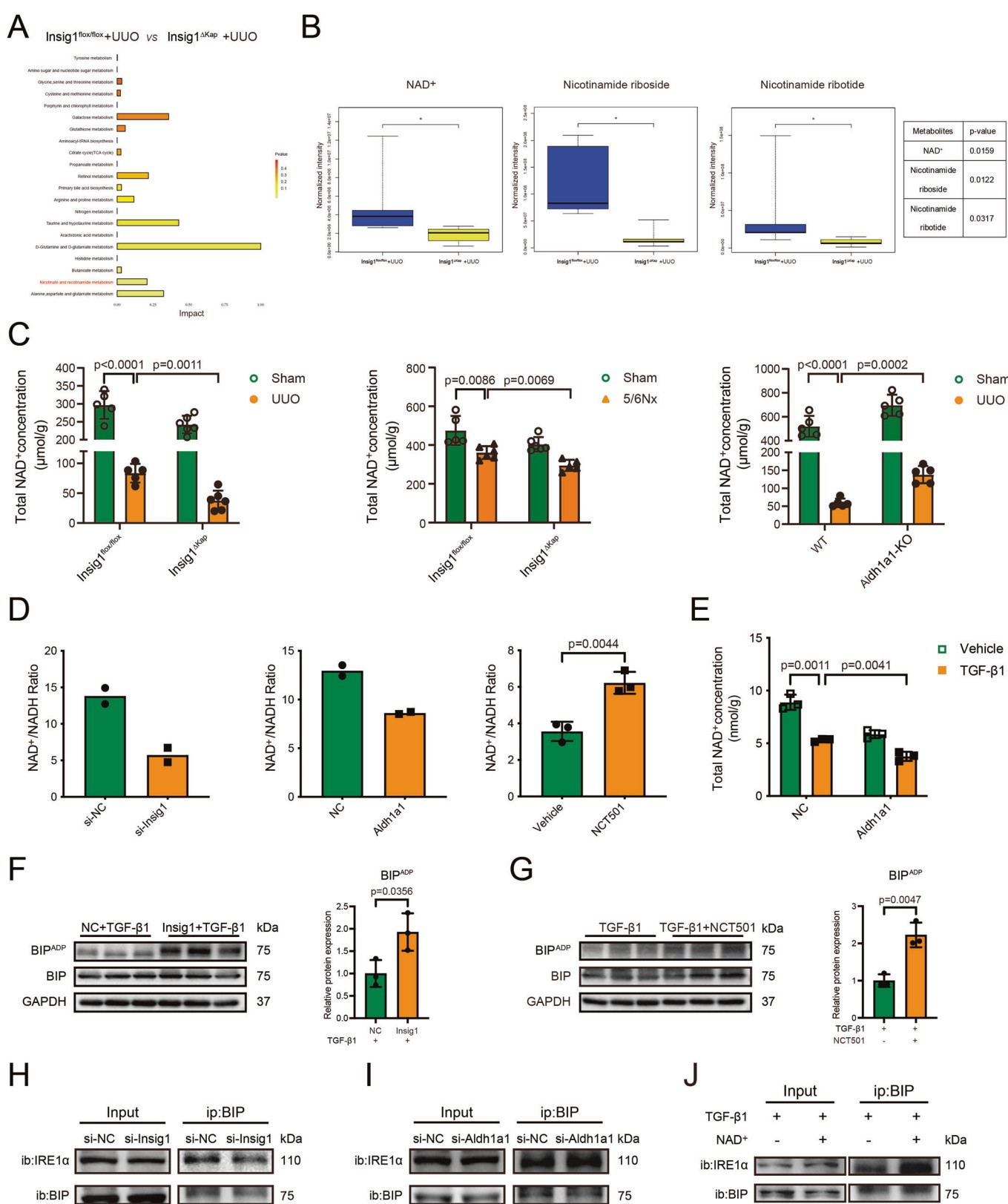

## Cell culture and treatment

TKPTS (American Type Culture Collection (ATCC), Manassas, VA, USA, CRL-3361) and HEK293T cell line (ATCC, CRL-3216) were maintained in DMEM-F12 medium containing 10% fetal bovine serum (FBS). HK2 cells (obtained from FuHeng Biology Cell Bank, Shanghai, China, FH0228) were maintained in RPMI 1640 medium containing 10% FBS. All cells were cultured in a humidified incubator

◀ **Figure 8.  Insig1/Aldh1a1 activation mitigated renal fibrosis by increasing NAD⁺ levels.**

(A) KEGG analysis of the Insig1$^{flox/flox}$ + UUO and Insig1$^{\Delta Kap}$ + UUO groups. (B) The concentrations of NAD⁺, nicotinamide riboside, and nicotinamide ribotide were analyzed in the Insig1$^{flox/flox}$ + UUO and Insig1$^{\Delta Kap}$ + UUO groups ($n = 5$ in each group, biological replicates; the upper and lower edges of the box are the upper and lower quartiles, the center line of the box is the median, the box whisker boundary is the maximum and minimum, and there are no outiers). (C) NAD⁺ concentration was measured in the kidneys of Insig1$^{\Delta Kap}$ mice after UUO ($n = 5$ in Insig1$^{flox/flox}$ group; $n = 6$ in Insig1$^{\Delta Kap}$ group, biological replicates) or 5/6 Nx ($n = 5$–6 in each group, biological replicates), and Aldh1a1-KO mice after UUO ($n = 5$ in each group, biological replicates). (D) The NAD⁺/NADH ratio was detected in Insig1-silenced, Aldh1a1-overexpressing, or NCT501-treated TKPTS cells ($n = 2$–3 in each group, biological replicates). (E) NAD⁺ concentration was detected in Aldh1a1-overexpressing TKPTS cells stimulated by TGF-β1 ($n = 3$ in each group, biological replicates). (F, G) Representative immunoblotted bands and quantification of ADP-ribosylated BIP in Insig1-overexpressing or NCT501-treated TKPTS cells stimulated by TGF-β1 ($n = 3$ in each group, biological replicates). (H–J) Co-IP analysis of the IRE1α and BIP binding in Insig1-silenced or Aldh1a1-silenced or NAD⁺-treated TKPTS cells. Data information: In (B), Data are represented as mean ± SD. Student's *t* test or Mann–Whitney test. (C–G) Data are represented as mean ± SD. Student's *t* test. Source data are available online for this figure.

(with 5% $CO_2$ and 37 °C). The cells were specifically grown in DMEM or RPMI 1640 medium plus 10% FBS, penicillin (100 U/ml), and streptomycin (100 μg/ml). All cell lines were authenticated by short tandem repeat (STR) DNA profiling and were verified to be mycoplasma-free. The cells were transfected with Insig1 or Aldh1a1 plasmids or siRNA using PolyJet reagent (SL100688, SignaGen, Maryland, USA) or Lipo 3000 (L3000015, ThermoFisher, IL, USA) and then stimulated with TGF-β1 (10 ng/ml, 240-B-002, R&D Systems, Minneapolis, MN, USA) for 24 h. For the pharmaceutical intervention combined with TGF-β1, cells were pretreated with NCT501 (60 μM) or nicardipine (3 μM) for 2 h before stimulation with TGF-β1 (10 ng/ml) or vehicle for 24 h. The information on plasmids or siRNA is listed in Table EV2.

### Dual-luciferase reporter assay

First, pGL4.19-mAldh1a1 promoter constructs were cotransfected into HEK293T cells with Insig1, SREBP1 plasmids or si-Insig1. Second, the pGL4.19-mAldh1a1 promoter constructs were transfected into HEK293T cells with si-Insig1 or SREBP1 (1–480 bp), followed by treatment with either vehicle or fatostatin. Cells were assayed using a Dual-Luciferase Reporter Assay System (Promega, E1910). Firefly luciferase activity was normalized to the corresponding Renilla luciferase activity.

### ChIP assay

ChIP assays were conducted using a Simple ChIP Plus Enzymatic Chromatin IP Kit (CST, 9006) according to the manufacturer's protocol. After the ChIP DNA sequences were obtained, qPCR was performed using primers which are listed in Table EV3) to amplify genomic regions containing Aldh1a1-/SREBP1-binding motifs. Enrichment for gene loci was reported as the fold difference in +enrichment of the Aldh1a1/SREBP1 antibody relative to that of the control IgG antibody.

### RNA isolation and quantitative real-time PCR (qRT-PCR)

The total RNA was extracted from kidney cortexes and cells using TRIzol (Invitrogen, 15596026) according to the manufacturer's protocol. We reverse-transcribed total RNA (1 μg) into cDNAs using the PrimeScript™ Reverse Transcriptase System (TaKaRa, RR036A). qRT-PCR was performed on the LightCycler 96 Real-time PCR System (Roche, Basel, Switzerland) with the AceQ qPCR SYBR Green Master Mix (Q111-02, Vazyme, Nanjing, China). The

expression of each gene was normalized to that of the control gene glyceraldehyde-3-phosphate dehydrogenase (GAPDH). The primers used for PCR amplification are listed in Table EV3.

### Western blotting

Frozen kidney cortex tissue and cells were added to two hundred microliters of lysis buffer (supplemented with 1×protease inhibitor cocktail (04693132001, Roche) and incubated on ice for 30 min. The protein concentration was measured using the BCA method (23227, ThermoFisher), and 50 μg of total protein was subjected to Western blotting analysis with standard methods using primary antibodies against FN (Abcam, ab2413, 1:1000), α-SMA (Abcam, ab5694, 1:1000), COL-I (Bioss, bs-10423R, 1:1000), CTGF (Proteintech, 23936-1-AP, 1:1000), α-TUBULIN (Proteintech, 11224-1-AP), INSIG1 (Abcam, ab70784, 1:800; Santa Cruz, sc-390504, 1:800; Proteintech, 55282-1-AP, 1:1000), IRE1α (Proteintech, 27528-1-AP, 1:800), ALDH1A1 (Proteintech, 15910-1-AP, 1:800), BIP (Santa Cruz, sc-13539, 1:800), SREBP1 (Santa Cruz, sc-13551, 1:800), ALDH2 (Bioworld, BS6268, 1:1000), ALDH1B1 (Bioworld, BS5604, 1:1000), ALDH1A2 (Bioworld, BS8957, 1:1000), E-CADHERIN (CST, 14472, 1:1000), VIMENTIN (CST, 5741, 1:1000), LAMIN B1 (Proteintech, 12987-1-AP, 1:800), GAPDH (Proteintech, 60004-1-Ig, 1:1000), followed by the addition of HRP-labeled secondary antibodies (CST, 7076 and 7074, 1:2000). Densitometric analysis was performed with Image J software (NIH, Bethesda, MD, USA).

### NAD⁺ measurement

The concentration of NAD⁺ in kidney tissues and cells were measured by the LC-MS/MS method and the ELISA kit (Abcam, ab65348) following the manufacturer's instructions. The absorbance of each sample was analyzed with a Synergy-H1 fluorimeter from Bio-Tek (Winooski, VT, USA).

### RNA sequencing analysis

Total RNA samples were obtained from isolated renal tubule tissues (Insig1$^{\Delta Kap}$ and Insig1$^{flox/flox}$ mice). Transcriptome sequencing was carried out by APTBIO (Shanghai, China), which performed RNA preparation, library creation, sequence analysis, and identification. Differential expression and KEGG analyses were performed using the DESeq2 R package (1.16.1). A *P* value < 0.05 and | log2fold change | >1 were considered to be significant.

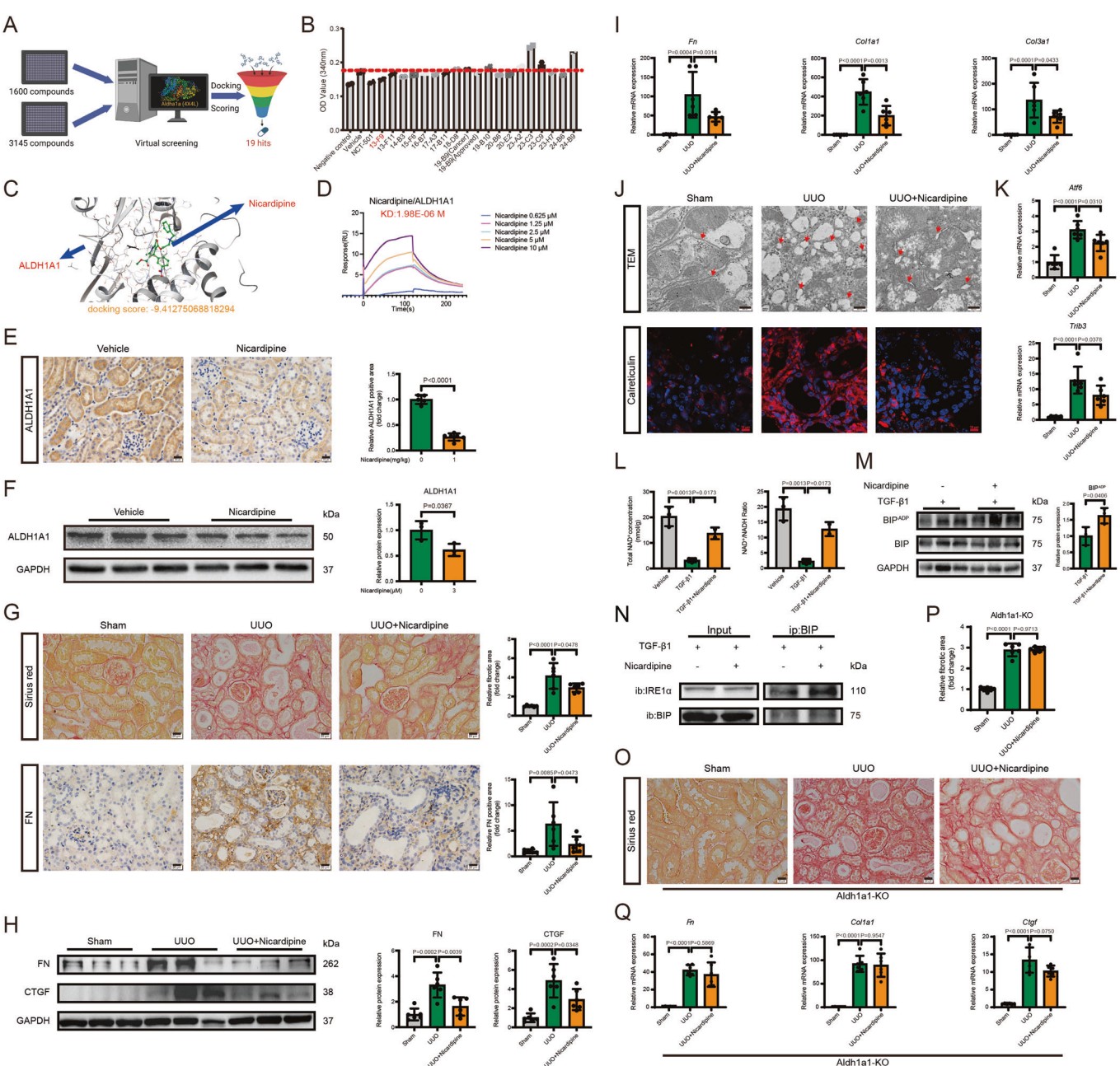

**Figure 9. Identification of a novel human Aldh1a1 inhibitor that alleviated UUO-induced CKD in a Aldh1a1-dependent manner.**

(A) Diagram showing the virtual screening of previously unknown human Aldh1a1 inhibitors. (B) Nineteen selected compounds were further detected using in vitro enzymatic experiments ($n = 2$ in each group, biological replicates). (C) Molecular simulation of the hAldh1a1 (PDB: 4X4L)–nicardipine complex. (D) Biacore assay of the binding strength of hAldh1a1 and nicardipine. (E) Representative images and quantification of IHC (ALDH1A1) in the kidneys (scale bars, 20 μm; $n = 5$ in vehicle group, $n = 6$ in nicardipine group, biological replicates). (F) Representative immunoblot bands and quantification of ALDH1A1 in nicardipine-treated TKPTS cells ($n = 3$). (G) Representative images and quantification of Sirius red staining and IHC (FN) in mouse kidneys (scale bars, 20 μm; $n = 6$, in each group, biological replicates). (H) Representative immunoblot bands and quantification of FN and CTGF in nicardipine-treated UUO kidneys ($n = 6$, in each group, biological replicates). (I) qPCR analysis of *Fn*, *Col1a1*, and *Col3a1* in nicardipine-treated UUO kidneys ($n = 6$, in each group, biological replicates). (J) Representative electron micrographs of ER in PTCs (scale bars, 500 nm; red arrows denote expanded ER) and calreticulin in the mouse kidneys (scale bars, 10 μm; $n = 3$ in each group, biological replicates). (K) qPCR analysis of *Atf6* and *Trib3* in nicardipine-treated UUO kidneys ($n = 6$, in each group, biological replicates). (L) NAD$^+$ concentration and the NAD$^+$/NADH ratio were detected in nicardipine-treated TKPTS cells stimulated by TGF-β1 ($n = 2$–3 in each group, biological replicates). (M) Representative immunoblot bands and quantification of ADP-ribosylated BIP in nicardipine-treated TKPTS cells stimulated by TGF-β1 ($n = 3$ in each group, biological replicates). (N) Co-IP analysis of the binding of IRE1α and BIP in nicardipine-treated TKPTS cells stimulated by TGF-β1. (O, P) Representative images and quantification of Sirius red staining in mouse kidneys (scale bars, 20 μm; $n = 6$, in each group, biological replicates). (Q) qPCR analysis of *Fn*, *Col1a1*, and *Ctgf* in UUO-induced kidneys of Aldh1a1-KO mice treated with or without nicardipine ($n = 6$, in each group, biological replicates). Data information: (E, F, M) Data are represented as mean ± SD. Student's $t$ test. (G–I, K, L, P, Q) Data are represented as mean ± SD. One-way ANOVA. Source data are available online for this figure.

**The paper explained**

**Problem**

Chronic kidney disease (CKD) is a leading cause of death, however few effective treatments are available. Renal fibrosis is a common outcome of CKD and leads to the final and irreversible loss of renal function. Profibrotic proximal tubules (PTs) are identified as a unique phenotype of PTCs in renal fibrosis. Controlling the process of renal fibrosis requires understanding how to manage the S1 subset's branch to the S3 subset rather than to the profibrotic PT subset. Insulin-induced gene 1 (Insig1) is one of the branch-dependent genes involved in controlling this process, although its role in renal fibrosis is unknown.

**Results**

We discovered the pathogenic effect of PTC-specific Insig1 deficiency on renal fibrosis in vivo and in vitro. Overexpression of Insig1 profoundly inhibited renal fibrosis. Insig1 deletion boosted SREBP1 nuclear localization, increasing Aldh1a1 transcriptional activity, causing excessive $NAD^+$ consumption and ER enlargement, and accelerating renal fibrosis. More importantly, we identified nicardipine as a selective inhibitor of Aldh1a1-restored $NAD^+$ and ER homeostasis, which attenuated renal fibrosis.

**Impact**

These findings not only provide evidence that maintaining the homeostasis of tubular Insig1 is a novel mechanism in mediating CKD, but also identify nicardipine as a specific Aldh1a1 antagonist for treating CKD.

## Untargeted metabolomics analysis

The metabolomics analysis was carried out by BioNovogene (Suzhou, China). Kidney tissues were harvested and processed following the standard method by BioNovogene. Chromatographic separation was carried out using an ACQUITY UPLC® HSS T3-equipped thermo vanquish system. The MS studies were conducted using a Thermo Q Exactive Focus mass spectrometer in positive and negative modes, with spray voltages of 3.5 kV and $-2.5$ kV, respectively. Bioinformatics techniques were used to analyze differentially abundant metabolites.

## Statistical analysis

All experimental data are presented as the mean ± SD. Statistical analyses were performed with GraphPad Prism (version 9.1.0, GraphPad Software, La Jolla, CA, USA). Statistically significant differences were determined via a parametric statistical analysis (Student's *t* test or one-way ANOVA) and non-parametric statistical analysis (Mann–Whitney test). Pearson correlation analysis was also used. A value of $P < 0.05$ was considered to be significant.

## For more information

See Kidney Interactive Transcriptomics: http://www.humphreyslab.com/SingleCell/.

## Data availability

Our RNA sequencing data have been deposited in the NCBI BioProject database under accession code PRJNA989392. The metabolomics data have been deposited in the Metabolights database under accession code MTBLS9985. All other data generated or analyzed during this study are included in this published article (and its supplementary information files). Source data are provided with this paper.

The source data of this paper are collected in the following database record: biostudies:S-SCDT-10_1038-S44321-024-00081-7.

## Peer review information

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

## Acknowledgements

This work was supported by grants from the National Natural Science Foundation of China (no. 82170753, 81974084). The authors thank the Chemical Biology Core Facility of the Center for Excellence in Molecular Cell Science for technical support in molecular docking and computational analysis.

## Author contributions

**Shumin Li**: Data curation; Formal analysis; Validation; Investigation; Visualization; Methodology; Writing—original draft. **Jun Qin**: Data curation; Formal analysis; Visualization; Methodology. **Yingying Zhao**: Data curation; Investigation; Visualization; Methodology. **Jiali Wang**: Data curation; Investigation; Methodology. **Songming Huang**: Resources; Funding acquisition; Validation; Investigation. **Xiaowen Yu**: Conceptualization; Formal analysis; Funding acquisition; Validation; Investigation; Methodology; Writing—review and editing.

Source data underlying figure panels in this paper may have individual authorship assigned. Where available, figure panel/source data authorship is listed in the following database record: biostudies:S-SCDT-10_1038-S44321-024-00081-7.

## Disclosure and competing interests statement

The authors declare no competing interests.

# Expanded View Figures

**Figure EV1.   Re-analysis of scRNA-seq dataset GSE182256 and the expression of ER stress indexes and Insig1 in the CKD model.**

(A) UMAP dimension Reduction showing 28 distinct cell types in Control and UUO kidneys identified by unsupervised clustering. GEC, glomerular endothelial cell; Endo, endothelial; Podo, podocyte; ALOH, ascending loop of Henle; DCT, distal convoluted tubule; CNT, connecting tubule; CD PC, collecting duct principal cell; A-IC, α intercalated cell; B-IC, β intercalated cell; Trans-IC, transitional intercalated cell; Neutro, neutrophil; Mono, monocyte; Macro, macrophage; Baso, basophil; B Lymph, B lymphocyte; Prolif Ly, proliferating lymphocyte; Prolif PT, proliferating PT. (B) Bubble plots showing the expression of cell cluster marker genes in UUO kidney. (C) Bubble plots showing the expression of PTCs cluster marker genes in UUO kidney. (D) qPCR analysis of *Perk*, *Trib3*, and *Atf6* in the kidneys of UUO model ($n = 5$, in each group, biological replicates). (E) qPCR analysis of *Atf4* in the kidneys of 5/6 Nx model ($n = 6$ in Sham group; $n = 5$ in 5/6 Nx group, biological replicates). (F) The violin plot showed the Insig1 expression levels in each type of normal kidney cells ($n = 6$ in Control group; $n = 2$ in UUO group). (G) The Insig1 expression changes in each PT subsets in UUO and Control groups. (H, I) Representative images and quantification of IHC (INSIG1) in kidneys of UUO model ($n = 5$ in Sham group; $n = 7$ in UUO group, biological replicates; scale bars, 20 μm). (J) qPCR analysis of *Insig1* in the kidneys of UUO model ($n = 6$ in Sham group; $n = 7$ in UUO group, biological replicates). (K) Representative immunoblot bands and quantification of INSIG1 in the kidneys of UUO model ($n = 6$, in each group, biological replicates). Data information: In (D, I), Data are represented as mean ± SD. Student's *t* test. (E, J, K) Data are represented as mean ± SD. Mann–Whitney test. Source data are available online for this figure.

▶

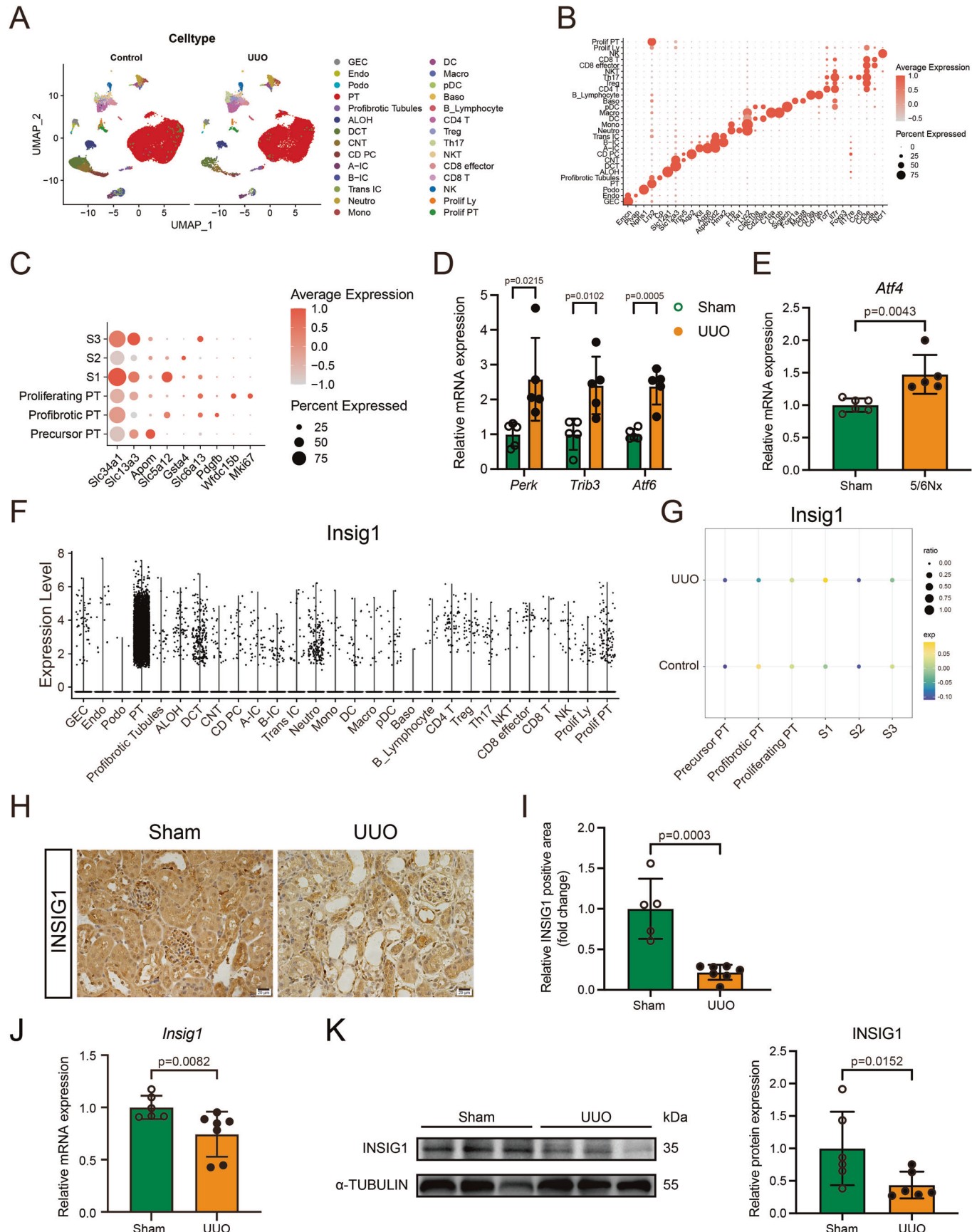

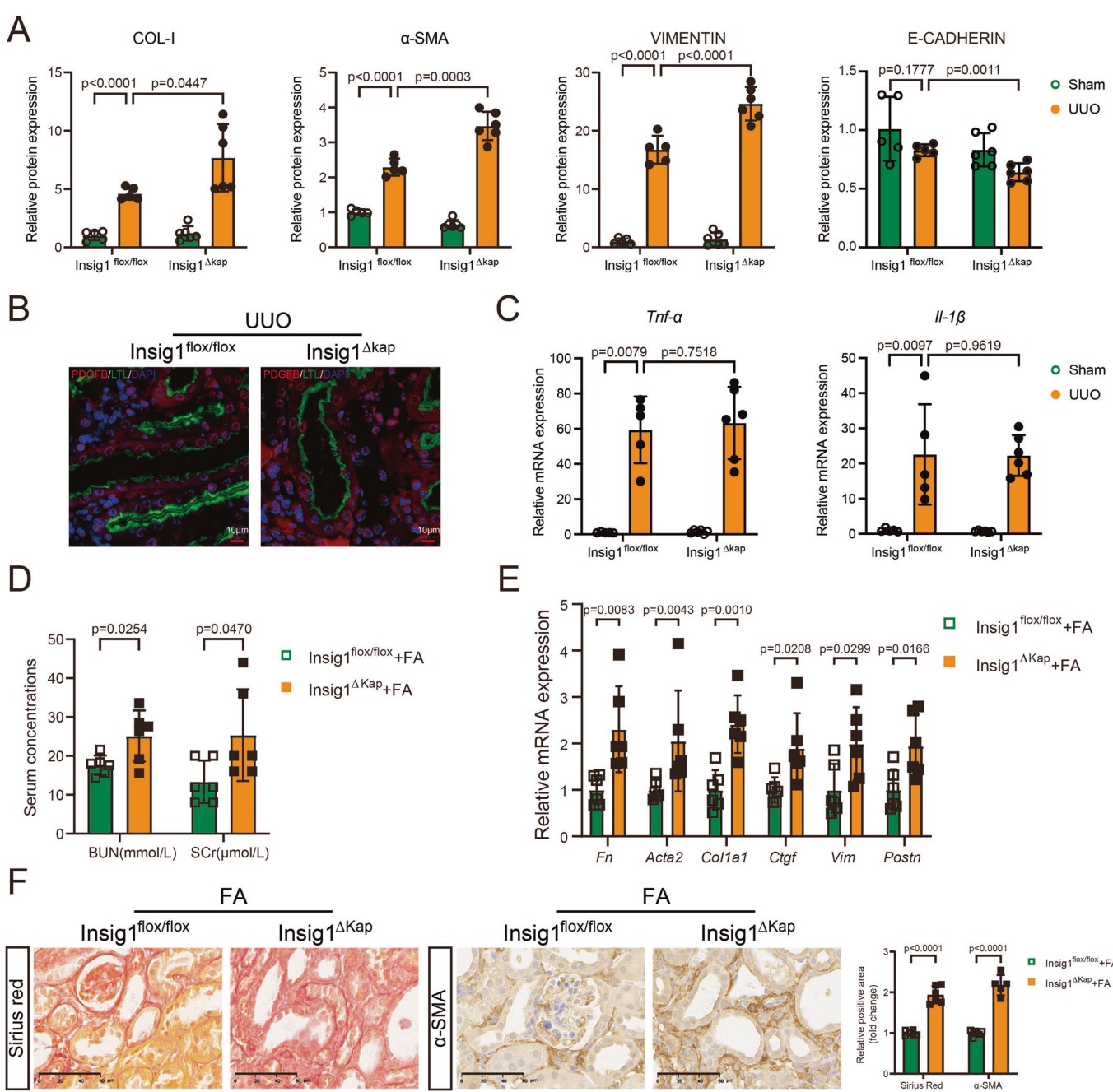

**Figure EV2. Insig1 loss in PTCs exacerbated UUO-induced and FA-induced kidney fibrosis.**

(A) Quantification of COL-I, α-SMA, VIMENTIN, and E-CADHERIN in Fig. 2F ($n = 5$ in Insig1$^{flox/flox}$ group and Insig1$^{flox/flox}$ + UUO group; $n = 6$ in Insig1$^{ΔKap}$ group and Insig1$^{ΔKap}$ + UUO group, biological replicates). (B) Representative images of PDGFB staining in Insig1$^{flox/flox}$ + UUO and Insig1$^{ΔKap}$ + UUO groups ($n = 3$ in each group, biological replicates; scale bars, 10 μm; red: PDGFB; green: LTL; blue: DAPI). (C) qPCR analysis of *Tnf-α* and *Il-1β* in the kidneys of Insig1$^{flox/flox}$ and Insig1$^{ΔKap}$ mice after UUO ($n = 5$ in Insig1$^{flox/flox}$ group and Insig1$^{flox/flox}$ + UUO group; $n = 6$ in Insig1$^{ΔKap}$ group and Insig1$^{ΔKap}$ + UUO group, biological replicates). (D) SCr and BUN concentrations were measured in FA-treated Insig1$^{flox/flox}$ and Insig1$^{ΔKap}$ mice ($n = 6$ in each group, biological replicates). (E) qPCR analysis of *Fn, Acta2, Col1a1, Ctgf, Vim*, and *Postn* in the kidneys of FA-treated Insig1$^{flox/flox}$ and Insig1$^{ΔKap}$ mice ($n = 6$ in each group, biological replicates). (F) Representative images and quantification of Sirius red staining and IHC (α-SMA) staining in mouse kidneys (scale bars, 60 μm; $n = 5$–6 in each group, biological replicates). Data information: In (A, D, F), Data are represented as mean ± SD. Student's *t* test. (C, E) Data are represented as mean ± SD. Student's *t* test or Mann–Whitney test. Source data are available online for this figure.

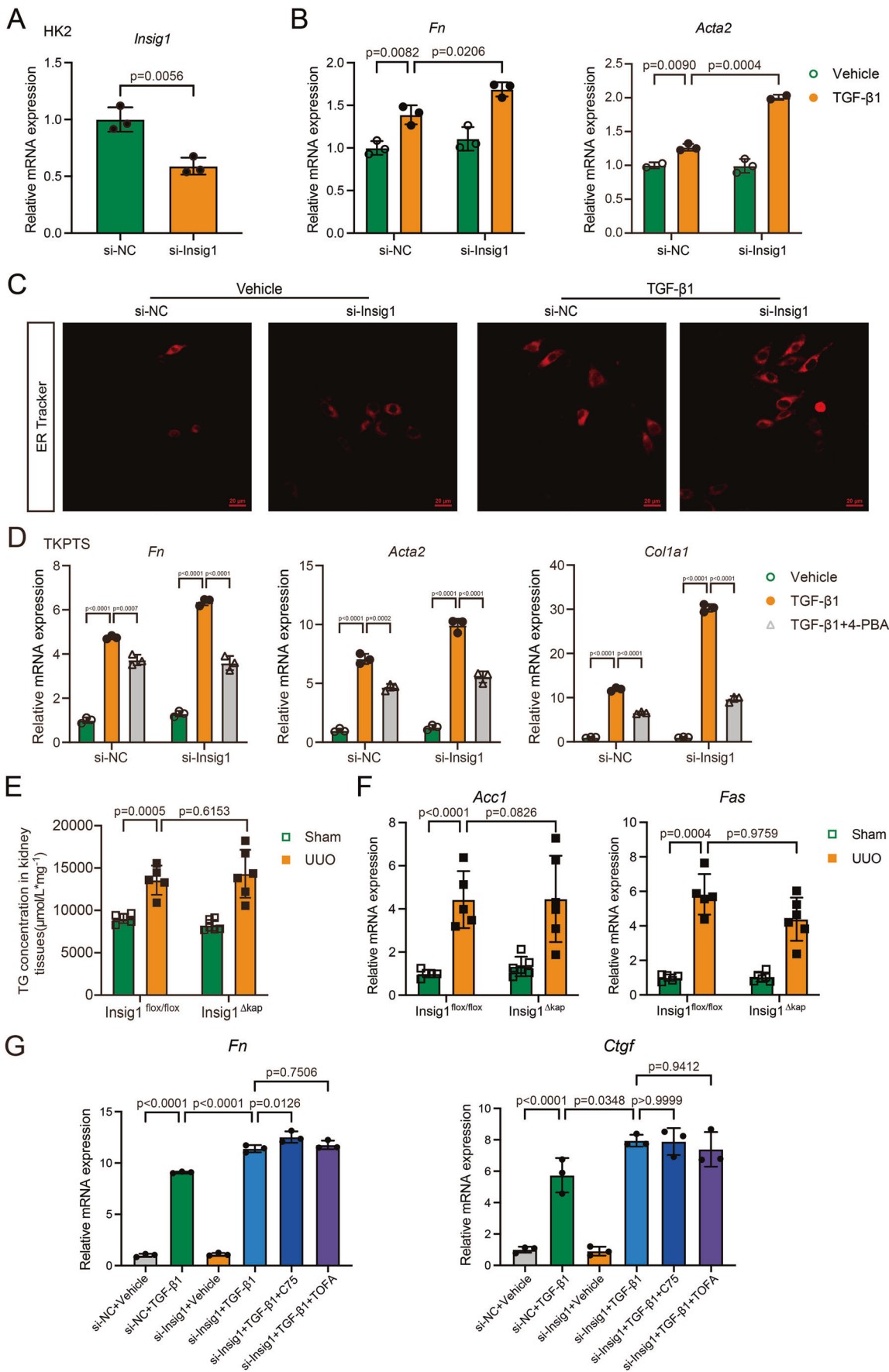

◀

**Figure EV3. The profibrotic effect of Insig1 deletion in PTCs was not be correlated with lipogenic activity.**

(A) qPCR analysis of *Insig1* in Insig1 silencing HK2 cells ($n = 3$ in each group, biological replicates). (B) qPCR analysis of *Fn* and *Acta2* in Insig1 silencing HK2 cells stimulated by TGF-β1 ($n = 2$–3 in each group, biological replicates). (C) Representative images of ER tracker staining in Insig1 silencing HK2 cells stimulated by TGF-β1 (scale bars, 20 μm; $n = 3$ in each group, biological replicates). (D) qPCR analysis of *Fn, Acta2* and *Col1a1* in Insig1 silencing TKPTS cells stimulated by TGF-β1 combined with 4-PBA ($n = 3$ in each group, biological replicates). (E) TG concentration was measured in the kidneys of Insig1$^{flox/flox}$ and Insig1$^{\Delta Kap}$ mice after UUO ($n = 5$ in Insig1$^{flox/flox}$ group and Insig1$^{flox/flox}$ + UUO group; $n = 6$ in Insig1$^{\Delta Kap}$ group and Insig1$^{\Delta Kap}$ + UUO group, biological replicates). (F) qPCR analysis of *Acc1* and *Fas* in the kidneys of Insig1$^{flox/flox}$ and Insig1$^{\Delta Kap}$ mice subjected to UUO ($n = 5$ in Insig1$^{flox/flox}$ group and Insig1$^{flox/flox}$ + UUO group; $n = 6$ in Insig1$^{\Delta Kap}$ group and Insig1$^{\Delta Kap}$ + UUO group, biological replicates). (G) qPCR analysis of *Fn* and *Ctgf* in Insig1 silencing TKPTS cells stimulated by TGF-β1 combined with C75 or TOFA ($n = 3$ in each group, biological replicates). Data information: In (A, B, E, F), Data are represented as mean ± SD. Student's *t* test. (D, G) Data are represented as mean ± SD. One-way ANOVA. Source data are available online for this figure.

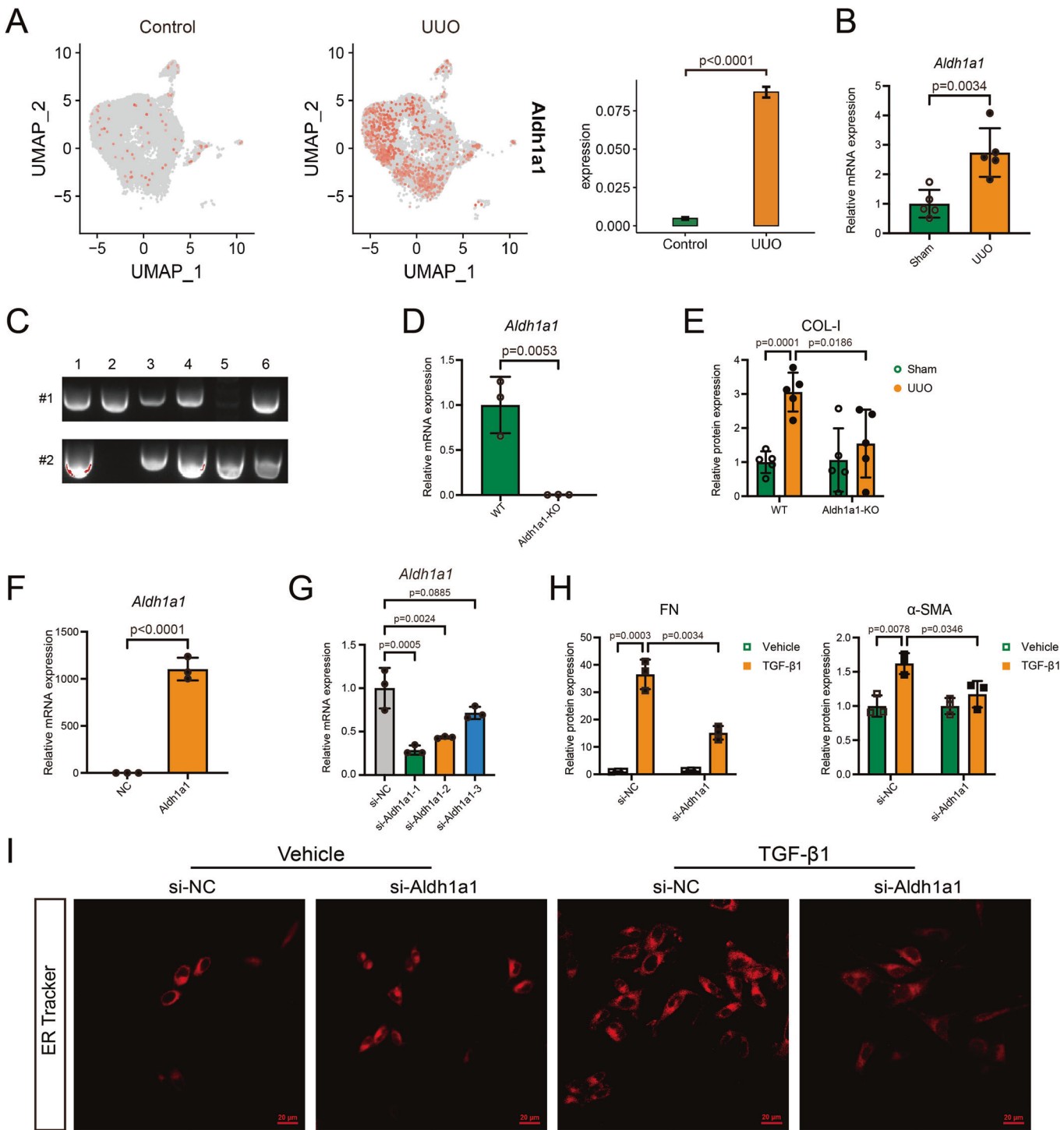

**Figure EV4. The expression of Aldh1a1 in UUO model and Aldh1a1 silencing ameliorated TGF-β1-induced fibrotic responses and ER stress in vitro.**

(A) UMAP dimension Reduction showing Aldh1a1 expression in Control and UUO kidneys ($n = 6$ in Control group; $n = 2$ in UUO group). (B) qPCR analysis of *Aldh1a1* in the kidneys of UUO model ($n = 5$ in each group, biological replicates). (C) PCR analysis of *Aldh1a1* for the genetic identification of Aldh1a1-KO mice. (HET: 1, 3, 4, 6; HOM: 2; WT: 5). (D) qPCR analysis of *Aldh1a1* in the kidneys of WT and Aldh1a1-KO mice ($n = 3$ in each group, biological replicates). (E) Quantification of COL-I in Fig. 7E ($n = 5$ in each group, biological replicates). (F) qPCR analysis of *Aldh1a1* in Aldh1a1 overexpressed TKPTS cells ($n = 3$ in each group, biological replicates). (G) qPCR analysis of *Aldh1a1* in Aldh1a1 silencing TKPTS cells (si-Aldh1a1-1 was selected for additional experimentation) ($n = 3$ in each group, biological replicates). (H) Quantification of FN and α-SMA in Fig. 7L ($n = 3$ in each group, biological replicates). (I) Representative images of ER tracker staining in Aldh1a1 silencing TKPTS cells stimulated by TGF-β1 (scale bars, 20 μm; $n = 3$ in each group, biological replicates). Data information: In (B, D, E, F, H), Data are represented as mean ± SD. Student's *t* test. (G) Data are represented as mean ± SD. One-way ANOVA. Source data are available online for this figure.

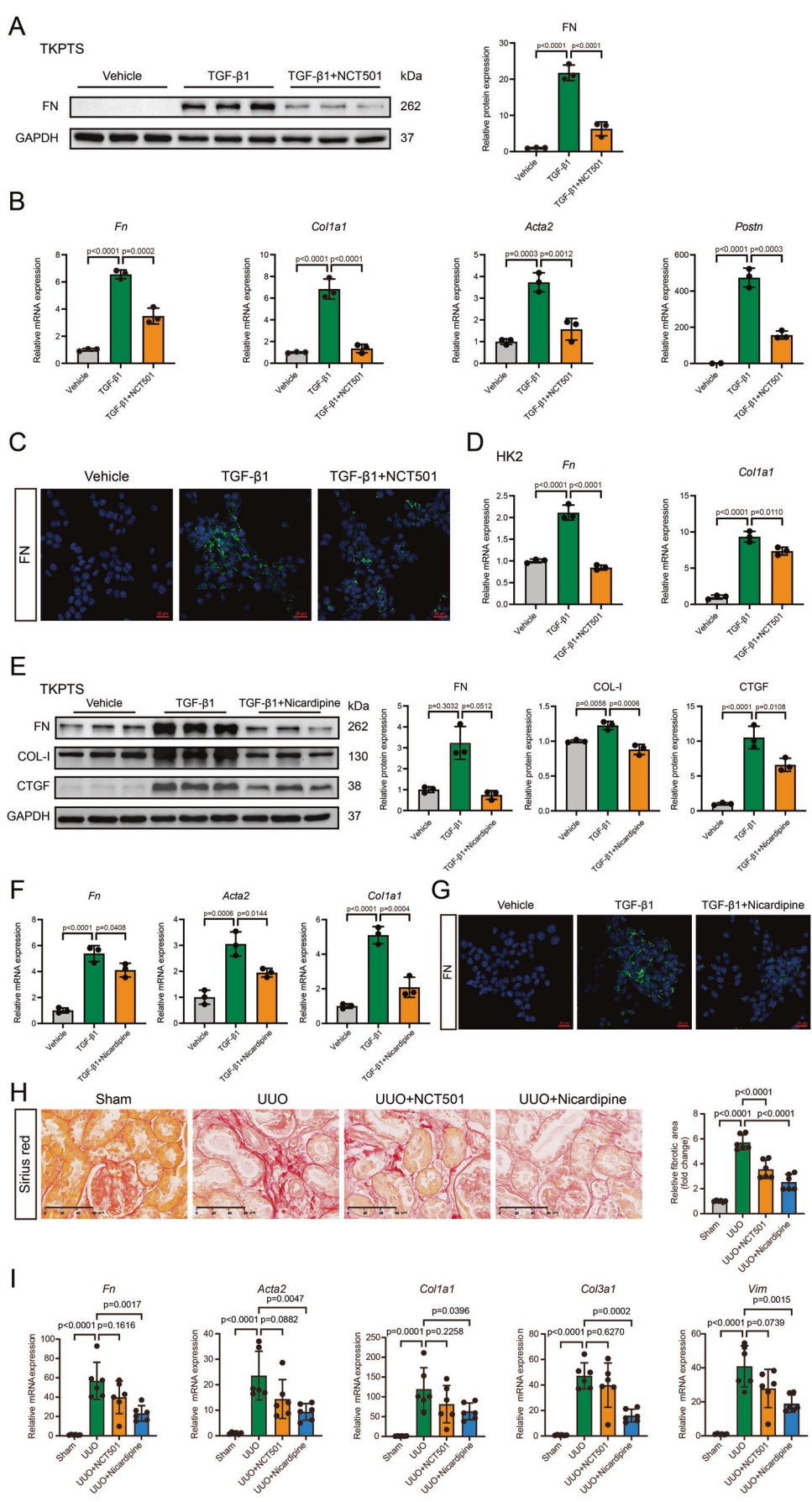

**Figure EV5. NCT501 and nicardipine improved renal fibrosis in vivo and in vitro.**

(A) Representative immunoblot bands and quantification of FN in TGF-β1-treated TKPTS cells stimulated with NCT501 ($n = 3$ in each group, biological replicates). (B) qPCR analysis of *Fn, Acta2, Col1a1* and *Postn* in TGF-β1-treated TKPTS cells stimulated with NCT501 ($n = 2$–3 in each group, biological replicates). (C) Representative images of FN staining in TGF-β1-treated TKPTS cells stimulated with NCT501 (scale bars, 20 μm; $n = 3$ in each group, biological replicates). (D) qPCR analysis of *Fn* and *Col1a1* in TGF-β1-treated HK2 cells stimulated with NCT501 ($n = 3$ in each group, biological replicates). (E) Representative immunoblot bands and quantification of FN, COL-I, and CTGF in TGF-β1-treated TKPTS cells with or without nicardipine treatment ($n = 3$ in each group, biological replicates). (F) qPCR analysis for *Fn, Acta2,* and *Col1a1* in TGF-β1-treated TKPTS cells with or without nicardipine treatment ($n = 3$ in each group, biological replicates). (G) Representative images of FN staining in TGF-β1-treated TKPTS cells with or without nicardipine treatment (scale bars, 20 μm; $n = 3$ in each group, biological replicates). (H) Representative images and quantification of Sirius red staining in mouse kidneys (scale bars, 60 μm; $n = 6$ in each group, biological replicates). (I) qPCR analysis for *Fn, Acta2, Col1a1, Col3a1* and *Vim* in mouse kidneys ($n = 6$ in each group, biological replicates). Data information: Data are represented as mean ± SD. One-way ANOVA. Source data are available online for this figure.

