## [Peer Review File · EMBO Molecular Medicine]

Tubular insulin-induced gene 1 deficiency promotes NAD⁺ consumption and exacerbates kidney fibrosis

Shumin Li, Jun Qin, Yingying Zhao, Jiali Wang, Songming Huang, and Xiaowen Yu

Corresponding authors: *xiaowen yu (yuxiaowen@njmu.edu.cn)* , *Songming Huang (smhuang@njmu.edu.cn)*

Review Timeline:

Submission Date:	28th Dec 23
Editorial Decision:	5th Jan 24
Appeal:	12th Jan 24
Editorial Decision:	15th Feb 24
Revision Received:	18th Apr 24
Editorial Decision:	24th Apr 24
Revision Received:	27th Apr 24
Accepted:	13th May 24

Editor: Poonam Bheda

Transaction Report:

4th Jan 2024

Decision on your manuscript EMM-2023-19208

Dear Dr. Yu,

Thank you for submitting your manuscript "Tubular insulin-induced gene 1 deficiency promotes NAD⁺ consumption and exacerbates kidney fibrosis" to EMBO Molecular Medicine. I have now carefully read your manuscript and discussed it with my colleagues. I regret to say that we all agree that the manuscript is not well suited for publication in EMBO Molecular Medicine and therefore have decided not to proceed with peer review.

In this study, you identify Insig1 as an ER stress related gene that is downregulated in patients with chronic kidney disease and you show that loss of Insig1 in proximal tubule cells in mice promotes renal fibrosis upon UO. We appreciate that you report that mechanistically Insig1 expression is regulated by Aldh1a1, and that you identify a new Aldh1a1 inhibitor, nicardipine, that reduces fibrosis in mice and the expression of fibrotic genes in cells. However, we feel that the conceptual advance and clinical significance presented in the work remains limited given that there are several other Aldh1a1 inhibitors and that a therapeutic improvement as a result of nicardipine administration over e.g. NCT-501 has not been demonstrated.

Thus, we are unfortunately not persuaded that your manuscript provides the sort of clinical impact and advance we would expect in an EMBO Molecular Medicine article. I am sorry that I could not bring better news this time but hope that this negative decision does not prevent you from considering our journal for the publication of your future studies.

Yours sincerely,

Poonam Bheda, PhD
Scientific Editor
EMBO Molecular Medicine

As a service to authors, EMBO provides authors with the possibility to transfer a manuscript that one journal cannot offer to publish to another EMBO publication. The full manuscript and if applicable, reviewers reports are automatically sent to the receiving journal to allow for fast handling and a prompt decision on your manuscript. For more details of this service, and to transfer your manuscript to another EMBO title please click on Link Not Available

Dear Poonam Bheda,

Thank you for reviewing my manuscript. I regret that my research has not been able to get your approval. However, I would like to explain the questions you raised about my research.

Firstly, we discovered that Insig1 deletion in PTCs boosted SREBP1 nuclear localization, increased Aldh1a1 transcriptional activity, caused excessive NAD⁺ consumption and ER enlargement, and accelerated renal fibrosis, not “Insig1 expression is regulated by Aldh1a1”.

Secondly, the equilibrium dissociation constant (KD) value of Aldh1a1 with nicardipine was 1.98E-06 M, indicating strong binding between nicardipine and Aldh1a1, while the KD value of Aldh1a1 with NCT501 was 3.31E-05 M. What's more, Nicardipine is effective in the treatment of blood pressure in clinical, but here, we found that it effectively attenuated CKD, expanding its clinical indications to treatment of CKD patients with hypertension, which demonstrated the clinical significance in the work. Furthermore, the therapeutic improvement of nicardipine administration over e.g. NCT-501 had been demonstrated pathological staining (Specific statistics were not provided). Later, we can compare the therapeutic improvement of two drugs in a series of experiments.

We still sincerely hope that this work will attract your interest. We appreciate you considering it for publication in the *EMBO Molecular Medicine* journal, and we hope to hear from you soon.

Sincerely yours,

Songming Huang & Xiaowen Yu MD, Ph.D.

Professor of Medicine, Nanjing Medical University.

于晓文

yuxiaowen@njmu.edu.cn

15th Feb 2024

Dear Dr. Yu,

Thank you for the submission of your manuscript to EMBO Molecular Medicine. We have now received feedback from the three reviewers who agreed to evaluate your manuscript. As you will see from the reports below, the referees acknowledge the interest of the study and are overall supportive of your work; however they also comment on multiple aspects of the manuscript that should be strengthened in a revision.

In particular, improvements in the presentation and figures are necessary. In addition, the revisions should include a comparison of nicardipine with NCT501 (Reviewer 1), analyze your transcriptomics data for differential expression of Insig1 (Reviewer 2), and investigate the effect of Insig PTC-cKO on E-cadherin and vimentin expression (Reviewer 3). In a cross-consultation session between the editor and reviewers, there was discussion as to whether renal glomeruli need to be included in the Sirius Red staining images as the study primarily focuses on observing renal interstitial fibrosis induced by UUO and partial nephrectomy, rather than glomerular damage. Therefore for this concern, you may include experimental evidence or include a justification to Reviewer 2 as to why this may not be necessary.

Addressing the reviewers' concerns in full in a point-by-point response will be necessary for further considering the manuscript in our journal, and acceptance of the manuscript will entail a second round of review. EMBO Molecular Medicine encourages a single round of revision only and therefore, acceptance or rejection of the manuscript will depend on the completeness of your responses included in the next, final version of the manuscript. For this reason, and to save you from any frustrations in the end, I would strongly advise against returning an incomplete revision. If you would like to discuss further the points raised by the referees, I am available to do so via email or video. Let me know if you are interested in this option.

We are expecting your revised manuscript within three months, if you anticipate any delay, please contact us. When submitting your revised manuscript, please carefully review the instructions that follow below. We perform an initial quality control of all revised manuscripts before re-review; failure to include requested items will delay the evaluation of your revision.

We require:

- 1) A .docx formatted version of the manuscript text (including legends for main figures, EV figures and tables). Please make sure that the changes are highlighted to be clearly visible.
- 2) Individual production quality figure files as .eps, .tif, .jpg (one file per figure). For guidance, download the 'Figure Guide PDF' (<https://www.embopress.org/page/journal/17574684/authorguide#figureformat>).
- 3) At EMBO Press we ask authors to provide source data for the main figures. Our source data coordinator will contact you to discuss which figure panels we would need source data for and will also provide you with helpful tips on how to upload and organize the files.
- 4) A .docx formatted letter INCLUDING the reviewers' reports and your detailed point-by-point responses to their comments. As part of the EMBO Press transparent editorial process, the point-by-point response is part of the Review Process File (RPF), which will be published alongside your paper.
- 5) A complete author checklist, which you can download from our author guidelines (<https://www.embopress.org/page/journal/17574684/authorguide#submissionofrevisions>). Please insert information in the checklist that is also reflected in the manuscript. The completed author checklist will also be part of the RPF.
- 6) Please note that all corresponding authors are required to supply an ORCID ID for their name upon submission of a revised manuscript.
- 7) It is mandatory to include a 'Data Availability' section after the Materials and Methods. Before submitting your revision, primary datasets produced in this study need to be deposited in an appropriate public database, and the accession numbers and database listed under 'Data Availability'. Please remember to provide a reviewer password if the datasets are not yet public (see <https://www.embopress.org/page/journal/17574684/authorguide#dataavailability>).

In case you have no data that requires deposition in a public database, please state so in this section. Note that the Data Availability Section is restricted to new primary data that are part of this study. This study includes no data deposited in external repositories.

- 8) For data quantification: please specify the name of the statistical test used to generate error bars and P values, the number (n) of independent experiments (specify technical or biological replicates) underlying each data point and the test used to

calculate p-values in each figure legend. The figure legends should contain a basic description of n, P and the test applied. Graphs must include a description of the bars and the error bars (s.d., s.e.m.). Please provide exact p values.

13) Author contributions: CRediT has replaced the traditional author contributions section because it offers a systematic machine readable author contributions format that allows for more effective research assessment. Please remove the Authors Contributions from the manuscript and use the free text boxes beneath each contributing author's name in our system to add specific details on the author's contribution. More information is available in our guide to authors.

Please also suggest a striking image or visual abstract to illustrate your article as a PNG file 550 px wide x 300-600 px high. Share synopsis text and image, as well as eTOC:

Please note that these would be the final versions and changes during proofing are usually not allowed

16) As part of the EMBO Publications transparent editorial process initiative (see our policy here: https://www.embopress.org/transparent-process#Review_Process), EMBO Molecular Medicine will publish online a Peer Review File (PRF) to accompany accepted manuscripts.

In the event of acceptance, this file will be published in conjunction with your paper and will include the anonymous referee reports, your point-by-point response and all pertinent correspondence relating to the manuscript. Let us know whether you agree with the publication of the PRF and as here, if you want to remove or not any figures from it prior to publication. Please note that the Authors checklist will be published at the end of the RPF.

I look forward to receiving your revised manuscript.

Yours sincerely,

Poonam Bheda

Poonam Bheda, PhD
Scientific Editor
EMBO Molecular Medicine

***** Reviewer's comments *****

Referee #1 (Remarks for Author):

Li et al. carried out a comprehensive analysis of *Insig1* and *Aldh1a1* in kidney disease. In the kidney disease field, there is still an open question about which genes are in charge of renal fibrosis and how they realize that. In this study, their result demonstrated that *Insig1* deficiency enhanced the nuclear localization SREBP1 and the transcriptional activity of its target gene *Aldh1a1*, causing excessive NAD⁺ consumption, ER stress, and ultimate renal fibrosis. Overall, the results support their conclusions. In particular, the function of *Aldh1a1* in regulating NAD⁺ and further renal fibrosis provides important novelty. Due to some minor issues, I suggest giving a chance for minor revision. If these minor issues are solved, the manuscript will be acceptable for publication.

Minor issues:

1. The introduction section is too long, should be cut by approx. 10-20%.
2. Regarding human biopsies, 90% of samples were from children under 18 years, statements in the manuscript, such as "CKD patients" should be changed to "Children with kidney disease".
3. The method to isolate proximal tubules and fibroblasts in the kidney are not described.
4. Figure S5: for those not knowing lipid metabolism, please explain the chosen analysis to address whether *Insig1* effect is dependent on lipogenic activity.
5. Please explain how to choose *Aldh1a1* as a downstream target and exclude others?
6. What are the benefits of Nicardipine over NCT501 as a new *Aldh1a1* inhibitor?
7. English writing needs improvement.

Referee #2 (Remarks for Author):

There are obvious errors or lack of rigor in the paper graphics.

1. In Figure 1m, the IHC image should be white balanced to ensure consistent background brightness.
2. In Figure 2d, in the FN detection of *Insig1*Δ*Kap* mice with sham operation, in order to be focused on renal medulla, there should be focused on the field of view with glomeruli. In Figure 2f, the bands in western blot show that the expression of GAPDH in the groups of the sham and UO is uneven. It is recommended to be replaced.
3. In Figure 3a, b&c, the group of 5/6N treated *Insig1*Δ*Kap* mice, there might be a mouse which should be removed, because the renal function in each image was only one which was dispersed from other five ones. In Figure 3c, in the Sirius red detection of *Insig1*^{flox/flox} and *Insig1*Δ*Kap* mice treated with 5/6Nx, in order to be focused on renal medulla, there should be focused on the field of view with glomeruli.
4. In Figure 4b, the bands in western blot show that the expression of GAPDH in the groups of the sham and UO is uneven.
5. In Figure 5d, the representative images of FN staining in si-NC stimulated by TGF-β1, the number of cells is significantly less than other groups and needs to be adjusted to be consistent. In Figure 5i, the number of cells in both of the NC and *Insig1* overexpressed groups were significantly less than other groups. It seems like that there was cytotoxicity of the vehicle.
6. In Figure 6g, there should be corresponding statistical charts showing the obvious expression levels between the si-NC and si-*insig1*. The entire text should be drawn as a mechanism diagram rather than being divided into several parts, such as Figure 6i and Figure 8f.
7. In Figure 7a, the bands in western blot show that the expression of GAPDH in the groups were uneven. In Figure 7k, the number of cells in both of the NC and *Aldh1a1* overexpressed groups were significantly less than other groups. It seems like that there was cytotoxicity of the vehicle.

- 8.Regarding the differential expression of Insig1, is there any indication in the transcriptomic data of the UUO model.
 - 9.In Supplementary Figure S7a, the images in Masson staining should be white balanced to ensure consistent background brightness. In Supplementary Figure S7e and S11c, the images in three groups were not at the same magnification factor.
 - 10.In Supplementary Figure S9a, the bands of ip:BIP were unclear and fuzzy.
- In Supplementary Figure S10D, the bands in western blot show that the expression of GAPDH in the groups were uneven.

Referee #3 (Comments on Novelty/Model System for Author):

no

Referee #3 (Remarks for Author):

The authors established several fibrosis models by using several conditional knockout mouse models, demonstrating the crucial role of insig1 in renal fibrosis and its regulatory mechanisms. This research highlights new insights and potential strategies for the treatment of CKD. However, this reviewer has several concerns.

1. In Figure 2d, the Sirius Red staining of the insig1 flox/flox UUO group shows too little collagen deposition (bright red area). Typically, successful UUO results in severe fibrotic lesion. As demonstrated in Figure 7c, the Sirius Red staining and FN staining of the WT-UUO group showed a typical kidney fibrosis. Please select a more representative image.
2. In Figure 2f, the protein bands of GAPDH are visibly inconsistent, suggesting potential issues with the accuracy of protein loading for gel electrophoresis. Please clarify.
3. It is well established that the tubule epithelial-to-mesenchymal transition (EMT) plays a crucial role in the pathogenesis of renal fibrosis. Therefore, it would be better if the authors could investigate the effects of insig1 PTC-cKO on the expression of E-cadherin and vimentin in kidneys.

Responses to Reviewers:**Responses to reviewer #1:**

1) The introduction section is too long, should be cut by approx. 10-20%.

Response: Thanks for your valuable comments and suggestion. The introduction part has been shortened in the revised MS.

2) Regarding human biopsies, 90% of samples were from children under 18 years, statements in the manuscript, such as "CKD patients" should be changed to "Children with kidney disease".

Response: Thanks for your suggestion. We have changed it in the revised MS.

3) The method to isolate proximal tubules and fibroblasts in the kidney are not described.

Response: Thanks for your suggestion. We have added the description of proximal tubules and fibroblasts isolation in the Appendix Supplementary Methods section.

4) Figure S5: for those not knowing lipid metabolism, please explain the chosen analysis to address whether Insig1 effect is dependent on lipogenic activity.

Response: Thanks for your suggestion. We have added explanations in the revised MS.

The synthesis of TG is intimately linked to the activity of two key enzymes in lipid synthesis, ACC1 and FAS. We discovered that UUO-induced ACC1, FAS

expression, or TG content were not further increased by *Insig1* deletion in PTCs. In line with this finding, when *Insig1* was knocked down in TKPTS cells, TOFA (an inhibitor of acetyl-CoA carboxylase- α) and C75 (an inhibitor of fatty-acid) were unable to block the profibrotic effect of TGF- β 1. Thus, the profibrotic effect of *Insig1* deletion in PTCs did not appear to be linked to lipogenic activity.

5) Please explain how to choose *Aldh1a1* as a downstream target and exclude others?

Response: Thanks for your valuable comments and suggestion. We have added it in the Fig 6C and D.

6) What are the benefits of Nicardipine over NCT501 as a new *Aldh1a1* inhibitor?

Response: Thanks for your valuable comments and suggestion. We have deleted it in the revised MS.

(1) NCT501 is a potent and selective inhibitor of *Aldh1a1*, its half-life is still very short ($T_{1/2} < 1$ h) and clearance level was very high (CL = 98 mL/min/kg), which suggests that NCT501 is well absorbed and distributed but rapidly

metabolized and/or excreted. Therefore, further work focused on the improvement of the half-life and oral bioavailability is beneficial and worthy of done (Li et al, 2021).

(2) Nicardipine, a drug commonly used to treat hypertension in clinical, was found to be a specific inhibitor of Aldh1a1. Furthermore, it binds to Aldh1a1 with greater stability than NCT501.

(3) To evaluate the therapeutic improvement of nicardipine administration over NCT501, we compare the therapeutic effect of two drugs in the UO-induced CKD model. As shown in Fig EV5H-I, compared to NCT501, nicardipine treatment significantly reduced collagen deposition and the fibrotic indicators expression in the kidneys following UO surgery.

EV5H

EV5I

7) English writing needs improvement.

Response: Thanks for the comment. In this revised MS, we thoroughly checked and corrected the errors in typing and grammar.

Reference

Li B, Yang K, Liang D, Jiang C, Ma Z (2021) Discovery and development of selective aldehyde dehydrogenase 1A1 (ALDH1A1) inhibitors. *European journal of medicinal chemistry* 209: 112940

Responses to reviewer #2:

1) In Figure 1m, the IHC image should be white balanced to ensure consistent background brightness.

Response: Thanks for your suggestion. Following your suggestion, in the revised MS, we have repeated the immunohistochemical staining, using a pathological scanner to scan the slices for imaging, and changed the previous images in Fig 1N.

2) In Figure 2d, in the FN detection of *Insig1* Δ Kap mice with sham operation, in order to be focused on renal medulla, there should be focused on the field of

view with glomeruli. In Figure 2f, the bands in western blot show that the expression of GAPDH in the groups of the sham and UUO is uneven. It is recommended to be replaced.

Response: Thanks for your valuable comments. In Fig 2D, we have selected a more representative image in the revised MS. Even though there were the same amount of loaded proteins in each lane in Fig 2F, the expression of GAPDH is unequal due to the consideration of additional external factors. Following your comments, we re-examined the samples using WB method and changed all the bands in Fig 2F.

3) In Figure 3a, b&c, the group of 5/6nX treated Insig1^{ΔKap} mice, there might be a mouse which should be removed, because the renal function in each

image was only one which was dispersed from other five ones. In Figure 3c, in the Sirius red detection of *Insig1^{flox/flox}* and *Insig1^{ΔKap}* mice treated with 5/6Nx, in order to be focused on renal medulla, there should be focused on the field of view with glomeruli.

Response: Thanks for your valuable comments. Following your suggestion, a mouse with renal function was distributed among the other five has been removed in Fig 3. Additionally, in the revised MS, we have carried out the experiment again and re-selected the representative images in Fig 3C.

4) In Figure 4b, the bands in western blot show that the expression of GAPDH in the groups of the sham and UUO is uneven.

Response: Thanks for your valuable comments. We have carried out the

experiment again and changed it in the revised MS.

5) In Figure 5d, the representative images of FN staining in si-NC stimulated by TGF-β1, the number of cells is significantly less than other groups and needs to be adjusted to be consistent. In Figure 5i, the number of cells in both of the NC and Insig 1 overexpressed groups were significantly less than other groups. It seems like that there was cytotoxicity of the vehicle.

Response: Thanks for your valuable comments and suggestion. In the revised MS, we have re-selected the representative images in Fig 5D and I.

6) In Figure 6g, there should be corresponding statistical charts showing the

obvious expression levels between the si-NC and si-insig1. The entire text should be drawn as a mechanism diagram rather than being divided into several parts, such as Figure 6i and Figure 8f.

Response: Thanks for your suggestion. As shown in Fig 6I, we have added the statistical charts. We acknowledge your point of view and have removed the small diagrams in the revised MS. We divided them up to help the reader understand the results.

7) In Figure 7a, the bands in western blot show that the expression of GAPDH in the groups were uneven. In Figure 7k, the number of cells in both of the NC and Aldh1a1 overexpressed groups were significantly less than other groups. It seems like that there was cytotoxicity of the vehicle.

Response: Thanks for your important comments. In the revised MS, we have carried out the experiment again and changed the bands and representative images in Fig 7A and J.

7A

7J

8) Regarding the differential expression of *Insig1*, is there any indication in the transcriptomic data of the UUO model.

Response: Thanks for your important comments. Following your comments, we used an online database (GSE145053) to confirm the decrease in *Insig1* mRNA levels in UUO models (Fig 1M).

1M

9) In Supplementary Figure S7a, the images in Masson staining should be white balanced to ensure consistent background brightness. In Supplementary Figure S7e and S11c, the images in three groups were not at the same magnification factor.

Response: Thanks for your valuable comments and suggestion. Following the comment of reviewer #1, we compare the therapeutic effects of two drugs in the UUO-induced CKD model in order to evaluate the therapeutic improvement of nicardipine administration over NCT501 (Fig EV5H and I). Thus, we removed the Masson staining images of NCT501 in Figure S7a. Furthermore, the same rulers were automatically inserted by ZEN program, which surprised us even more with the data from Supplementary Figure S7e and S11c. We carried out the experiment again and re-selected the representative images (Fig EV5C and G).

EV5H

EV5I

EV5C

EV5G

10) In Supplementary Figure S9a, the bands of ip:BIP were unclear and fuzzy. In Supplementary Figure S10D, the bands in western blot show that the expression of GAPDH in the groups were uneven.

Response: Thanks for your comments. In the revised MS, as shown in Fig 8J and Appendix Figure S2E, the prior results were replaced after we collected the cell samples again and performed the WB experiment again.

8J

Appendix S2E

Responses to reviewer #3:

1) In Figure 2d, the Sirius Red staining of the *insig1* flox/flox UUO group shows too little collagen deposition (bright red area). Typically, successful UUO results in severe fibrotic lesion. As demonstrated in Figure 7c, the Sirius Red staining and FN staining of the WT-UUO group showed a typical kidney fibrosis. Please select a more representative image.

Response: Thanks for your valuable comments and suggestion. In the revised MS, we have re-selected the representative images in Fig 2D.

2) In Figure 2f, the protein bands of GAPDH are visibly inconsistent, suggesting potential issues with the accuracy of protein loading for gel electrophoresis. Please clarify.

Response: Thanks for your valuable comments and suggestion. Even though there were the same amount of loaded proteins in each lane in Fig 2F, the expression of GAPDH is unequal due to the consideration of additional

external factors. Following your comments, we re-examined the samples using WB method and changed all the bands.

2F

3) It is well established that the tubule epithelial-to-mesenchymal transition (EMT) plays a crucial role in the pathogenesis of renal fibrosis. Therefore, it would be better if the authors could investigate the effects of insig1 PTC-cKO on the expression of E-cadherin and vimentin in kidneys.

Response: Thanks for your valuable comments. Following your comments, we have investigated the effects of Insig1 PTC-cKO on the expression of E-cadherin and vimentin in the kidneys of Insig1^{flox/flox} and Insig1^{ΔKap} mice after UUO or FA treatment. Following UUO, the kidneys of Insig1^{ΔKap} animals displayed considerably lower expression of E-CADHERIN and significantly higher expression of VIMENTIN (Vim) (Fig 2E and F). Furthermore, after FA treatment, Insig1 deficiency in PTCs increased expression of Vimentin (Fig EV2E).

2E

2F

EV2E

24th Apr 2024

Dear Dr. Yu,

Thank you for the submission of your revised manuscript to EMBO Molecular Medicine. We have now received the enclosed reports from the referees that were asked to re-assess it. As you will see the reviewers are now globally supportive and I am pleased to inform you that we will be able to accept your manuscript pending the following final amendments and appropriate response to reviewers:

- 1) Please check the "Author Checklist" carefully and complete all relevant questions. Currently the author name and manuscript number are missing.
- 2) Abstract: Please check whether "Profibrotic proximal tubular (PT) were..." is correct. It seems like it should be "Profibrotic proximal tubules were...". Please also define PTCs in the abstract of the main text.
- 3) Data availability: Please ensure that the RNA-seq and metabolomics data are released, as they must be accessible prior to publication.
- 4) Author contributions: Please specify author contributions in our submission system. CRediT has replaced the traditional author contributions section because it offers a systematic machine-readable author contributions format that allows for more effective research assessment. You are encouraged to use the free text boxes beneath each contributing author's name to add specific details on the author's contribution. More information is available in our guide to authors: <https://www.embopress.org/page/journal/17574684/authorguide#authorshipguidelines>
- 5) In the Materials and Methods, please take care of the following:
 - Cell lines: Please be sure to include a sentence in the Materials and Methods as to whether or not the cell lines were recently tested for mycoplasma contamination.
 - Human research participants: Please be sure to include a sentence in the Materials and Methods as to whether the experiments conformed to the principles set out in the WMA Declaration of Helsinki and the Department of Health and Human Services Belmont Report.
- 6) Please place individual sections of the manuscript in the following order: Title page - Abstract & Keywords - Introduction - Results - Discussion - Materials & Methods - Data Availability - Acknowledgements - Disclosure and Competing Interests Statement - The Paper Explained - For More Information - References - Figure Legends - Expanded View Figure Legends.
- 7) For the figures and figure legends, please take care of the following:
 - Please note that a separate 'Data Information' section is required in the legends of figures 2d-f, j; 3a-d, f; 5b-j; 6d-k; 7c-f, i; 9g-i, k, o-q; EV 2a, c; EV 3a, c-g.
 - Please note that in the legend of figures 1d-e, the red arrows are incorrectly mentioned as red asterisks. This needs to be rectified.
 - Please define the annotated p values * in the legend of figure 8b; as appropriate.
 - Please indicate the statistical test used for data analysis in the legends of figures 1j; 6a-b, k.
 - Please note that the box plots need to be defined in terms of minima, maxima, centre, bounds of box and whiskers, and percentile in the legends of figures 1h; 8b.
 - Please note that information related to n is missing in the legends of figures 1i; 8b; EV 1f; EV 4a, e, h.
 - Please note that n=2 in figures 5a; 9b. Error bars cannot be calculated for measurements with n=2.
 - Please note that the error bars are not defined in the legends of figures 1i; 6k; 9q.
 - Please note that the red arrows are not defined in the legends of figures 2h; 3g; 7g; 9j. This needs to be rectified.
- Data citation:
 1. Please note that the (Ju et al, 2015) data citation does not refer to deposited experimental data.
 2. Please note that the data callouts in the text for (Doke et al, 2022), (Conway et al, 2020) and (Ju et al, 2015) data citation does not include "Data ref:" as a prefix.
- 8) Appendix file: Please move the Supplementary Methods in the Appendix to the Materials and Methods in the main manuscript file. Please also update the Appendix to include the manuscript title on the front page and page numbers for each figure. Please also reformat the red font to automatic/black font.
- 9) Synopsis:
 - Synopsis image: Please simplify the synopsis image and upload it as a high-resolution jpeg file 550 pixels wide x (250-400) pixels high. Currently the text in each cell will be difficult to read in a small format (please double check that it is readable after resizing).
 - Synopsis text: We would suggest rephrasing "worsened NAD+ consumption" as it is unclear whether this is increased or decreased. Our suggestion is the following: "Chronic kidney disease (CKD) mice with loss of Insulin-induced gene 1 (Insig1) in proximal tubule cells exhibit increased NAD+ consumption and worsened renal fibrosis, indicating that activation of Insig1 may offer a new approach to treating CKD."
 - Please check your synopsis text and image before submission with your revised manuscript. Please be aware that in the proof stage minor corrections only are allowed (e.g., typos).
- 10) As part of the EMBO Publications transparent editorial process initiative (see our policy here: https://www.embopress.org/transparent-process#Review_Process), EMBO Molecular Medicine will publish online a Peer Review File (PRF) to accompany accepted manuscripts. This file will be published in conjunction with your paper and will include the

anonymous referee reports, your point-by-point response and all pertinent correspondence relating to the manuscript. Let us know whether you agree with the publication of the PRF and as here, if you want to remove or not any figures from it prior to publication. Please note that the Authors checklist will be published at the end of the PRF.

11) Please provide a point-by-point letter INCLUDING my comments as well as the reviewer's reports and your detailed responses (as Word file).

I look forward to reading a new revised version of your manuscript as soon as possible.

Yours sincerely,

Poonam Bheda

Poonam Bheda, PhD
Scientific Editor
EMBO Molecular Medicine

*** Instructions to submit your revised manuscript ***

***** Reviewer's comments *****

Referee #1 (Remarks for Author):

I have no further comment, thank you.

Referee #2 (Comments on Novelty/Model System for Author):

This study maintained the homeostasis of tubular Insig1 is a novel mechanism in mediating CKD, but also identify nicardipine as a specific Aldh1a1 antagonist for treating CKD.

Referee #2 (Remarks for Author):

There were still obvious errors in the paper graphics :

1. In Figure 1m, the IHC image in health group, the renal tubules show significant swelling and occlusion, which is not the normal morphology of the renal tubules.

2. In Figure 2g, hollow and solid points, green and yellow bar charts represent different groups and should be distinguished from the Figure 2e. A hollow point should not represent Sham group and Insig 1 flox/flox+UUO group at the same time in Figure 2. In the following figures, there were the same questions. I suggest necessary modifications should be made.

Referee #3 (Comments on Novelty/Model System for Author):

No further comments.

Referee #3 (Remarks for Author):

No further comments.

Responses to Reviewers:**Responses to reviewer #1 :**

I have no further comment, thank you.

Response: We appreciate the valuable comments to improve the quality this manuscript and the kind support on this research work.

Responses to reviewer #2 :

1. In Figure 1n, the IHC image in health group, the renal tubules show significant swelling and occlusion, which is not the normal morphology of the renal tubules.

Response: Thanks for your valuable comments and suggestion. We have carefully examined the IHC images of the normal group, and found that some renal tubules were morphologically occluded and swollen. This was mainly because normal kidney tissue was taken from paracarcinoma. We have replaced the representative image in Fig 1N.

2. In Figure 2g, hollow and solid points, green and yellow bar charts represent different groups and should be distinguished from the Figure 2e. A hollow point should not represent Sham group and Insig 1 flox/flox+UUO group at the same time in Figure 2.

In the following figures, there was the same questions. I suggest necessary modifications should be made.

Response: Thanks for your valuable comments and suggestion. According to your suggestion, we have updated Fig 2G, Fig 3E, Fig 4F-H, Fig 6K, Fig 7F,K,M, Fig 8C,E, Fig EV2D-F, Fig EV3E,F, Fig EV4H.

Responses to reviewer #3 :

No further comments.

Response: We appreciate the valuable comments to improve the quality this manuscript and the kind support on this research work.

13th May 2024

Dear Dr. Yu,

We are pleased to inform you that your manuscript is accepted for publication and is now being sent to our publisher to be included in the next available issue of EMBO Molecular Medicine.

Yours sincerely,

Poonam Bheda, PhD
Scientific Editor
EMBO Molecular Medicine
